# Live-cell imaging reveals enhancer-dependent *Sox2* transcription in the absence of enhancer proximity

**Jeffrey M Alexander[1], Juan Guan[2], Bingkun Li[3], Lenka Maliskova[3], Michael Song[3,4], Yin Shen[3,4,5], Bo Huang[2,6,7], Stavros Lomvardas[8,9], Orion D Weiner[1,6]\***

[1]Cardiovascular Research Institute, University of California, San Francisco, San Francisco, United States; [2]Department of Pharmaceutical Chemistry, University of California, San Francisco, San Francisco, United States; [3]Institute for Human Genetics, University of California, San Francisco, San Francisco, United States; [4]Pharmaceutical Sciences and Pharmacogenomics Graduate Program, University of California, San Francisco, San Francisco, United States; [5]Department of Neurology, University of California, San Francisco, San Francisco, United States; [6]Department of Biochemistry and Biophysics, University of California, San Francisco, San Francisco, United States; [7]Chan Zuckerberg Biohub, San Francisco, United States; [8]Department of Biochemistry and Molecular Biophysics, Columbia University, New York City, United States; [9]Mortimer B Zuckerman Mind Brain and Behavior Institute, Columbia University, New York City, United States

**Abstract** Enhancers are important regulatory elements that can control gene activity across vast genetic distances. However, the underlying nature of this regulation remains obscured because it has been difficult to observe in living cells. Here, we visualize the spatial organization and transcriptional output of the key pluripotency regulator *Sox2* and its essential enhancer *Sox2* Control Region (SCR) in living embryonic stem cells (ESCs). We find that *Sox2* and SCR show no evidence of enhanced spatial proximity and that spatial dynamics of this pair is limited over tens of minutes. *Sox2* transcription occurs in short, intermittent bursts in ESCs and, intriguingly, we find this activity demonstrates no association with enhancer proximity, suggesting that direct enhancer-promoter contacts do not drive contemporaneous *Sox2* transcription. Our study establishes a framework for interrogation of enhancer function in living cells and supports an unexpected mechanism for enhancer control of *Sox2* expression that uncouples transcription from enhancer proximity.
DOI: https://doi.org/10.7554/eLife.41769.001

**\*For correspondence:**
orion.weiner@ucsf.edu

**Competing interests:** The authors declare that no competing interests exist.

## Introduction

Chromosomes are packaged and organized non-randomly within the mammalian nucleus. Emerging evidence suggests that 3D genome topology plays a fundamental role in genome control, including the regulation of gene expression programs (*Bickmore, 2013*; *Krijger and de Laat, 2016*; *Schwarzer and Spitz, 2014*). Within the nucleus, each chromosome occupies discrete chromosomal territories (*Cremer et al., 2006*). These territories are further structured into distinct compartments that separate active and repressive chromatin (*Lieberman-Aiden et al., 2009*; *Sexton et al., 2012*). At finer scales, chromosomes are partitioned into largely-invariant, sub-megabase sized topologically-associated domains (TADs), which break up the linear genome into interactive neighborhoods

(*Dixon et al., 2012*; *Nora et al., 2012*). Chromosomal contacts are disfavored across TAD boundaries. Thus, most cell-type specific contacts occur within TAD boundaries, and disruption of TAD architecture leads to dysregulation of gene expression (*Dowen et al., 2014*; *Gröschel et al., 2014*; *Guo et al., 2015*; *Lupiáñez et al., 2015*; *Narendra et al., 2015*; *Nora et al., 2017*).

Within this 3D framework, gene expression programs are established by non-coding regulatory enhancer elements. First discovered within a metazoan genome over three decades ago (*Banerji et al., 1983*), it is now predicted that greater than 300,000 enhancers are encoded in the human genome (*ENCODE Project Consortium, 2012*; *Zhu et al., 2013*). Enhancers demonstrate unique epigenetic markings, enriched for H3K4me1 and H3K27ac (*Creyghton et al., 2010*; *Heintzman et al., 2007*; *Rada-Iglesias et al., 2011*), and are highly accessible, as demonstrated by elevated DNase sensitivity and transposition susceptibility (*Boyle et al., 2008*; *Buenrostro et al., 2013*; *Thurman et al., 2012*). These features facilitate transcription factor occupancy, enrichment of co-activators such as p300 and Mediator, and transcription of non-coding enhancer RNAs (eRNAs), all of which play important roles in modulation of target gene expression (*Kim et al., 2015*; *Long et al., 2016*). Importantly, enhancer activity is highly specific across cell types (*Heintzman et al., 2009*; *ENCODE Project Consortium, 2012*; *Zhu et al., 2013*) and modulated during cellular differentiation (*Blum et al., 2012*; *Buecker et al., 2014*; *Huang et al., 2016*; *Wamstad et al., 2012*), and this activity correlates with nearby gene expression. Thus, enhancers are fundamental to achieving gene expression programs that orchestrate embryonic development and drive disease pathogenesis. Understanding the mechanism by which enhancers influence target genes is crucial to decode gene regulation.

The textbook model proposes that enhancers influence target gene promoters through protein-protein complexes and physical interaction mediated by a DNA loop (*Alberts et al., 2014*). Experimental support for this model comes primarily from numerous chromosome conformation capture (3C)-based studies that have identified enriched contacts between enhancer and promoter elements (*Jin et al., 2013*; *Li et al., 2012*; *Rao et al., 2014*; *Sanyal et al., 2012*; *Weintraub et al., 2017*) and recent observations that driving contacts between an enhancer-promoter pair is sufficient to augment gene expression (*Bartman et al., 2016*; *Deng et al., 2012*; *Deng et al., 2014*; *Morgan et al., 2017*). However, other observations fit this model poorly. For example, *sonic hedgehog* (*Shh*) enhancers that drive expression in the brain move further, rather than closer, to the *Shh* gene when activated (*Benabdallah et al., 2017*). Furthermore, in *Drosophila*, coupled reporter genes regulated by a shared enhancer nevertheless show coordinated transcriptional bursting, suggesting either that an enhancer can contact multiple genes at once or that contact can be decoupled from transcription (*Fukaya et al., 2016*; *Lim et al., 2018*). Super enhancers – clusters of enhancers that are highly enriched for coactivators like Mediator and BRD4 (*Lovén et al., 2013*; *Whyte et al., 2013*) – have been proposed to activate transcription through nucleation of activator droplets rather than stepwise assembly of transcription complexes (*Hnisz et al., 2017*), providing a possible mechanism for enhancer action at a distance, and recent imaging has provided support for this idea (*Cho et al., 2018*; *Sabari et al., 2018*). Thus, how distal elements communicate with and regulate gene promoters in living cells remains an open question.

Live-cell imaging represents a powerful approach to dissect chromatin architecture and gene regulation in the context of single cells to address these questions (*Chen et al., 2013*; *Chen et al., 2018*; *Germier et al., 2017*; *Gu et al., 2018*; *Lucas et al., 2014*). However, interrogation of both enhancer-gene spatial organization and real-time transcriptional activity of the regulated gene has not yet been realized in living mammalian cells. Here, we investigate the dynamic 3D organization and transcriptional activity of the *Sox2* gene and its distal enhancer *Sox2* Control Region (SCR) in mouse embryonic stem cells (ESCs) using live-cell microscopy.

We find that the *Sox2* promoter and SCR demonstrate similar spatial characteristics to non-regulatory regions in ESCs, while differentiation of ESCs leads to significant compaction throughout the *Sox2* region. Time-lapse microscopy revealed that individual loci explore only a fraction of their potential spatial range during the ~25 min imaging window, driving high cell-to-cell variability in *Sox2* locus conformation and *Sox2*/SCR encounters. Incorporation of an MS2 transcriptional reporter into the *Sox2* gene demonstrated that transcription occurs in intermittent bursts in ESCs but, surprisingly, showed no correlation with spatial proximity between the enhancer-promoter pair. Together, our findings establish the spatial and transcriptional characteristics of an essential pluripotency gene

and suggest an unconventional mechanism for enhancer control of *Sox2* expression that uncouples transcription from enhancer proximity.

## Results

### Engineering the endogenous *Sox2* locus to visualize locus organization in living Embryonic Stem Cells

To visualize discrete loci within the mammalian genome, we turned to the well-established genetic labeling method of incorporating repetitive arrays of exogenous operator sequences, an approach that has been extensively used to visualize chromosomal loci (*Belmont and Straight, 1998*; *Lucas et al., 2014*; *Marshall et al., 1997*; *Masui et al., 2011*; *Michaelis et al., 1997*; *Robinett et al., 1996*; *Roukos et al., 2013*). To independently visualize two regions of interest, we utilized the tetO/TetR system to visualize one chromosomal location. For the other chromosomal location, because of the reported issues using lacO/lacI in ESCs (*Lucas et al., 2014*; *Masui et al., 2011*), we developed a new tool based on the cuO/CymR pair. This is a repressor system from the bacteria *Pseudomonas putida* that is involved in cumate metabolism and has been previously used as a tool for inducible gene expression (*Mullick et al., 2006*). We opted to target these arrays to the mouse genome using a two-step genetic engineering strategy with bacteriophage integrases for two reasons (*Figure 1A*, see *Supplementary file 1* for protocol). First, repetitive sequences can be unstable during vector construction, making it advantageous to use generic targeting vectors portable between genomic loci. Second, we worried the repetitive arrays might recombine during genomic targeting using homologous recombination. To target the tetO/TetR and cuO/CymR labels to specific loci within the mouse genome, we first placed attP landing sites for the PhiC31 (*Raymond and Soriano, 2007*; *Thyagarajan et al., 2001*) and Bxb1 (*Xu et al., 2013*) integrase systems using CRISPR/Cas9 homology directed repair. We then integrated generic PhiC31 or Bxb1 targeting vectors bearing either the tetO array (224 repeats) or cuO array (144 repeats), respectively, at the corresponding landing sites through transient expression of the PhiC31 and Bxb1 integrases. This strategy was both modular in design and portable between genomic loci. To target two regions on the same chromosome, we used 129/Cast F1 hybrid ESCs, derived from crossing the 129 mouse strain to the divergent subspecies *Mus musculus castaneus*. This allowed us to limit editing to the 129 allele by using genetic polymorphisms between the two parental genomes to design allele-specific CRISPR guide RNAs.

We chose the murine *Sox2* locus as our genetic model. *Sox2* encodes a high-mobility group (HMG) DNA-binding transcription factor with important roles in embryonic development (*Kamachi and Kondoh, 2013*; *Lefebvre et al., 2007*; *Sarkar and Hochedlinger, 2013*), embryonic and adult neural progenitors (*Pevny and Nicolis, 2010*), and the progression of many forms of cancer (*Weina and Utikal, 2014*; *Wuebben and Rizzino, 2017*). *Sox2* also functions as an essential regulator of pluripotency, where it cooperates with other transcriptional regulators to maintain the pluripotency transcriptional program and keep embryonic stem cells in the undifferentiated state (*Chen et al., 2008a*; *Young, 2011*). *Sox2* resides in an isolated neighborhood on chromosome 3, as the sole protein-coding gene in a ~ 1.6 Mb region. Numerous regulatory elements that modulate *Sox2* expression have been identified in this neighborhood across amniotes (*Okamoto et al., 2015*; *Tomioka et al., 2002*; *Uchikawa et al., 2003*; *Zappone et al., 2000*). However, *Sox2* expression in mouse ESCs is controlled by a single, strong distal enhancer called the *Sox2* Control Region (*Li et al., 2014*; *Zhou et al., 2014*), which is robustly enriched with H3K27ac, DNase hypersensitivity, RNA Polymerase II (RNAP), CTCF, the cohesion subunit RAD21, and transcription factor occupancy (herein referred to as SCR, *Figure 1B*). Genetic ablation of SCR in ESCs leads to loss of *Sox2* expression in cis. Moreover, SCR maintains *Sox2* expression levels in the context of compound deletion of alternative *Sox2* enhancers, suggesting SCR is sufficient for *Sox2* regulation in ESCs (*Zhou et al., 2014*). Publicly available circularized chromosome conformation capture (4C) and HiC datasets reveal enriched contacts between SCR and the *Sox2* promoter region, suggesting that these enhancer-promoter interactions may play an important role in SCR function (*Figure 1B*).

We generated three distinct modified cell lines in 129/Cast F1 hybrid ESCs (*Figure 1B*, bottom) First, we labeled the *Sox2* promoter region and SCR by integrating the cuO array 8 kb centromeric to the *Sox2* TSS (*Sox2*-8C) and the tetO array approximately 5 kb telomeric to the SCR boundary

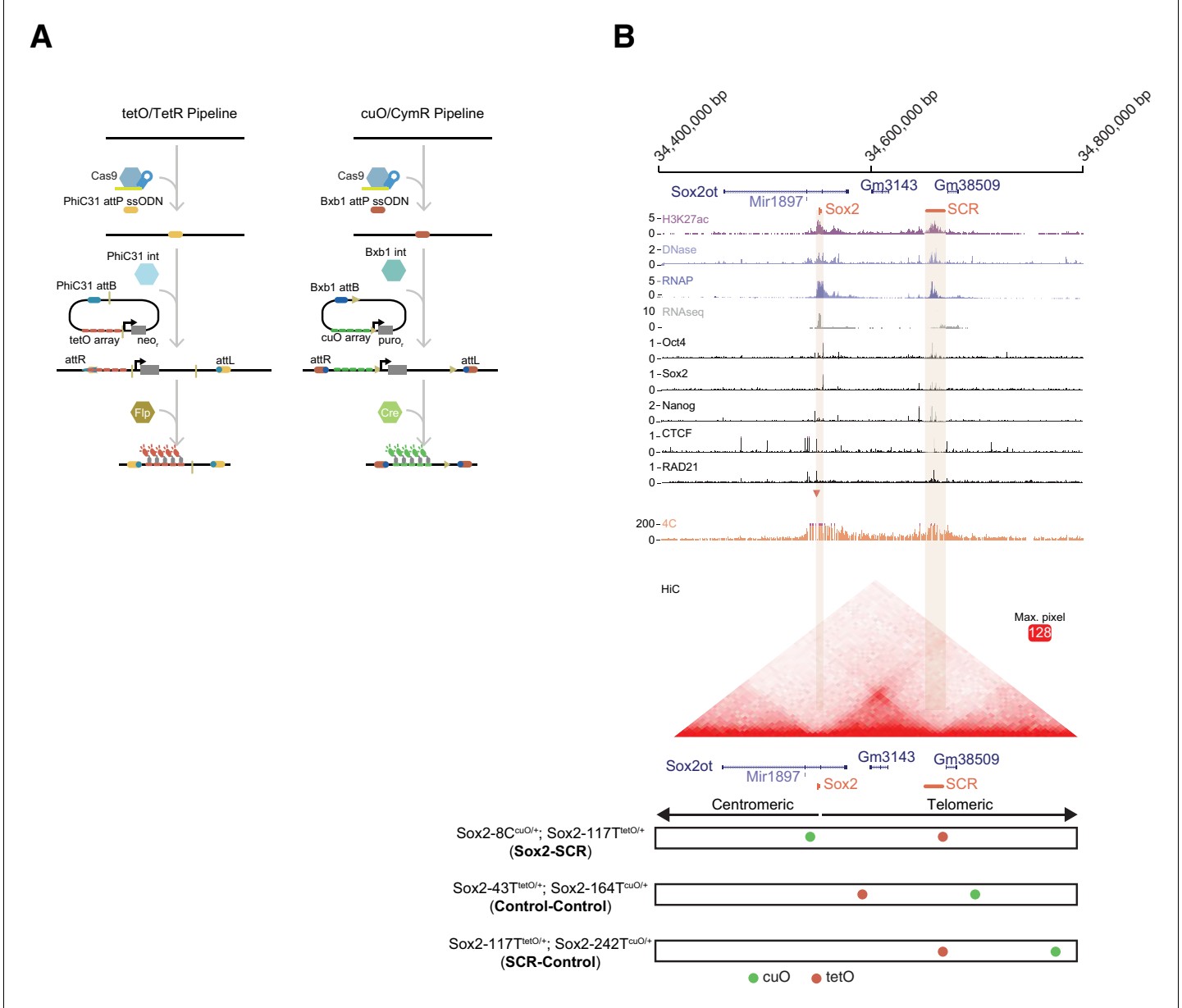

**Figure 1.** The *Sox2* Locus As a Model for Visualization of Enhancer-Promoter Regulation in Mouse Embryonic Stem Cells. (**A**) To visualize chromosome loci in living cells, we have used tetO/TetR and cuO/CymR genetic labels. Our pipeline for insertion of these labels into the mouse genome is shown. First, CRISPR-Cas9 is used to place an attP integrase landing site. Second, a targeting plasmid bearing the compatible attB sequence, the tetO or cuO array, and a selection cassette is introduced along the integrase (Int) to mediate site-specific integration. The selection cassette can then be subsequently removed by Cre/Flp recombinase. (**B**) The *Sox2* locus in mouse ESCs. Genomic browser tracks of epigenomic and expression data demonstrate high levels of histone acetylation, RNA polymerase II, and transcription factor (OCT4, SOX2, NANOG, CTCF) and cohesin (RAD21) occupancy at *Sox2* and the distal *Sox2* Control Region enhancer (tan boxes). Data from 4C and HiC experiments demonstrate chromosomal contacts at the *Sox2* locus. For 4C data, read density indicates contact frequency with a fixed position near the *Sox2* promoter (red triangle). Y-axis for browser tracks is reads per million. For HiC, all pairwise contact frequencies are shown using a heatmap. The intensity of each pixel represents the normalized number of contacts detected between a pair of loci. The maximum intensity is indicated in red square. At bottom, locations of the cuO- and tetO-arrays for the three cell lines utilized for this study. Sox2-8C$^{cuO/+}$; Sox2-117T$^{tetO/+}$ (Sox2-SCR) ESCs were used to track Sox2/SCR location. Two control lines, Sox2-43T$^{tetO/+}$; Sox2-164T$^{cuO/+}$ (Control-Control) and Sox2-117T$^{tetO/+}$; Sox2-242T$^{cuO/+}$ (SCR-Control) were analyzed for comparison. H3K27ac, RNA polymerase II (RNAP), and RNAseq data from GSE47949 (*Wamstad et al., 2012*); DNase data from GSE51336 (*Vierstra et al., 2014*); SOX2, OCT4, NANOG, CTCF data from GSE11431 (*Chen et al., 2008b*), and RAD21 data from GSE90994 (*Hansen et al., 2017*); 4C data from GSE72539 (*de Wit et al., 2015*); and HiChIP data from GSE96107 (*Bonev et al., 2017*).

DOI: https://doi.org/10.7554/eLife.41769.002

*Figure 1 continued on next page*

*Figure 1 continued*

The following figure supplements are available for figure 1:

**Figure supplement 1.** Characterization of Modified Embryonic Stem Cell Lines.
DOI: https://doi.org/10.7554/eLife.41769.003
**Figure supplement 2.** *Sox2* Expression Characterization for Modified Embryonic Stem Cell Lines.
DOI: https://doi.org/10.7554/eLife.41769.004
**Figure supplement 3.** *Sox2*-SCR Contacts Are Maintained in Modified Embryonic Stem Cell Lines.
DOI: https://doi.org/10.7554/eLife.41769.005

(i.e. 117 kb telomeric to *Sox2* TSS, *Sox2*-117T). We refer to this pair as *Sox2*-SCR. Secondly, we created two control ESC lines: one with two arbitrary loci labeled with cuO and tetO (*Sox2*-43T$^{tetO/+}$; *Sox2*-164T$^{cuO/+}$ or Control-Control) and a second where we labeled SCR along with a non-specific telomeric locus (*Sox2*-117T$^{tetO/+}$; *Sox2*-242T$^{cuO/+}$ or SCR-Control). In both cases, the genetic distance between labels was similar to that of *Sox2*-SCR. Both control pairs show low contact propensity in chromosome conformation capture data (*Figure 1B*). We verified the correct placement of the cuO and tetO labels for each locus using PCR with primers that span the unique recombination arms generated after plasmid integration (*Figure 1—figure supplement 1*, *Supplementary file 2,3*). We detected a similar *Sox2* expression ratio (129/Cast) using an allele-specific qPCR assay for modified cell lines compared to the parental ESCs, suggesting *Sox2* regulation is intact despite genetic alteration of the locus (Analysis of Variance, p=0.215, *Figure 1—figure supplement 2*). Furthermore, we found insertion of the cuO and tetO arrays within the *Sox2* locus did not disrupt *Sox2*-SCR contacts on the modified allele (*Figure 1—figure supplement 3*).

## Visualization of the *Sox2* region in ESCs reveals minimal evidence for *Sox2*/SCR Interactions

We were first interested in measuring the 3D distance between *Sox2* and the SCR enhancer in living ESCs. To this end, we stably coexpressed CymR-GFP and TetR-tdTomato (TetR-tdTom) fusion proteins in *Sox2*-SCR ESCs using ePiggyBac transposon-based gene delivery (*Lacoste et al., 2009*). This allowed for visualization of both the cuO and tetO arrays within the nucleus using live-cell fluorescence confocal microscopy. We confirmed that coexpression of CymR-GFP and TetR-tdTom did not significantly alter *Sox2* expression from the modified 129 allele by qPCR (*Figure 1—figure supplement 2*) and did not alter *Sox2*-SCR contacts by 4C (*Figure 1—figure supplement 3*). 3D time series of proliferating ESCs showed the majority of cells demonstrated a single, bright focus of CymR-GFP and TetR-tdTom in the ESC nucleus in close proximity. Many of these foci revealed the presence of two juxtaposed sister chromatids (*Video 1*). Because the overlapping signal from adjacent, identical arrays would degrade the resolution of our localization, we excluded these loci from our analysis and focused on cells demonstrating single, diffraction-limited spots for cuO and tetO, likely representing cells in the G1/early S phase of the cell cycle.

To investigate the distribution of *Sox2*/SCR distances, we determined the 3D position of cuO and tetO for each locus, assembled 3D tracks, and calculated 3D separation distances between the labels across time (*Figure 2A*, *Supplementary file 4*). 84% and 62% of our assembled tracks span the full time series (>75 frames) for cuO and tetO, respectively (*Figure 2—figure supplement 1*). By localization of fluorescent beads at a comparable signal-to-noise ratio, we estimate our localization precision in the X, Y, and Z dimensions to be 12 nm, 10 nm, and 36 nm, respectively, for cuO/CymR and 16 nm, 16 nm, 50 nm for tetO/TetR (*Figure 2—figure supplement 2*). Using fixed cells as an alternative method to estimate cuO/tetO

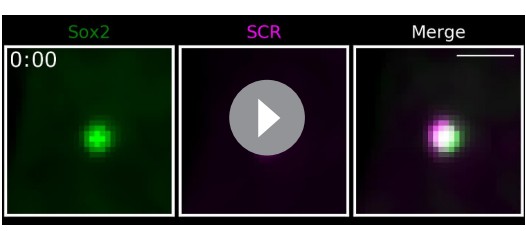

**Video 1.** Visualization of Sister Chromatids at *Sox2* Locus. Maximum-intensity Z projection of 3D confocal Z-stacks of cuO/CymR-GFP (left) and tetO/TetR-tdTom (middle) labeling the *Sox2* promoter region and SCR, respectively demonstrate two clear spots for the SCR label, suggesting cells in S/G2. These cells were excluded from analysis. Scale bar is 1 μm.
DOI: https://doi.org/10.7554/eLife.41769.006

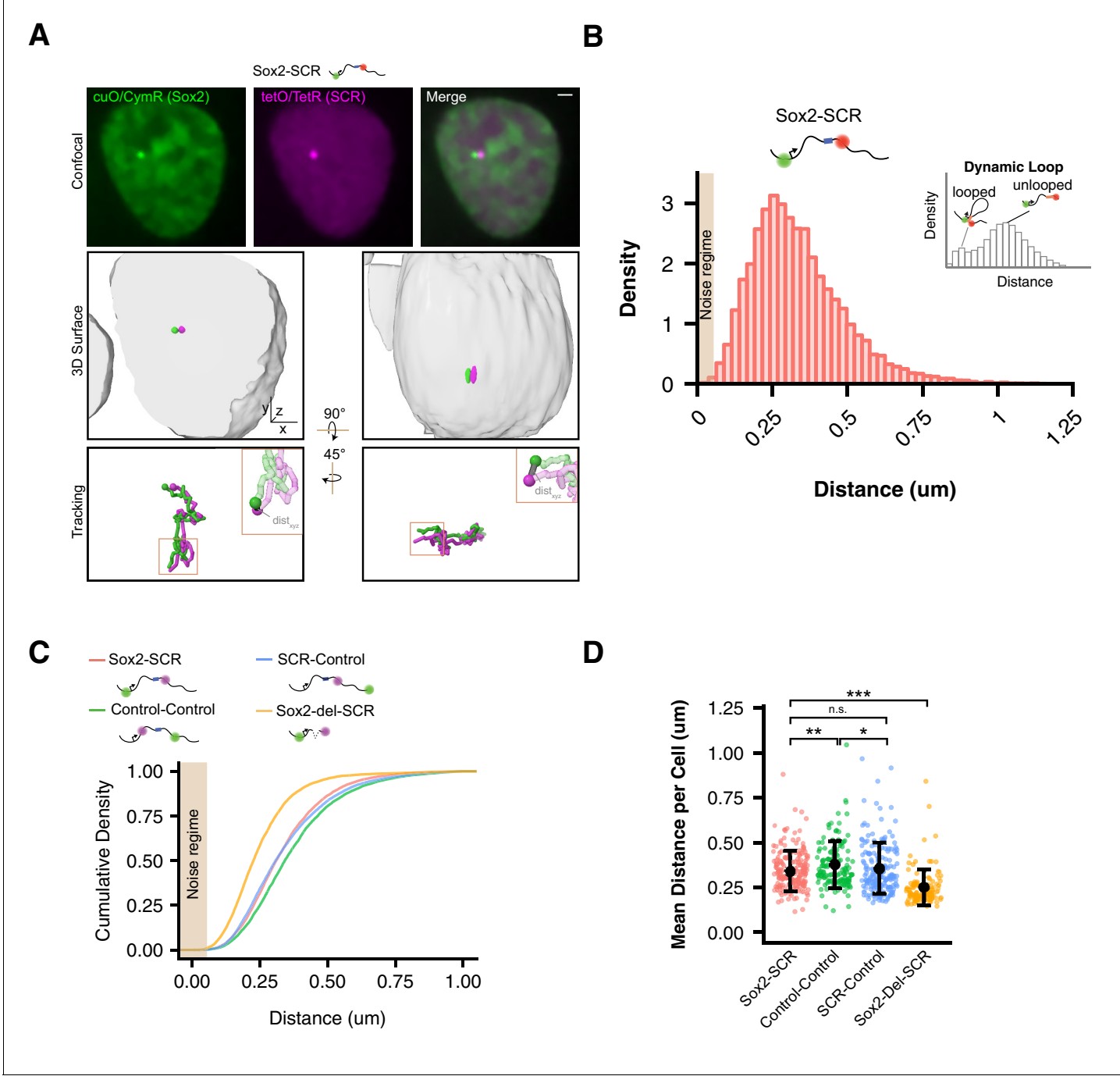

**Figure 2.** Visualization of the *Sox2* Region in ESCs Reveals Minimal Evidence for *Sox2*/SCR Interactions. (**A**) Top, confocal Z slices of CymR-GFP and TetR-tdTom in Sox2-SCR ESCs, labeling the Sox2 promoter and SCR region with bright puncta, respectively. Middle, 3D surface rendering of the ESC nucleus shown above. A single fluorescence channel was rendered white and transparent to outline the nucleus, and GFP and tdTom surfaces were rendered with high threshold to highlight the cuO and tetO arrays, respectively. Bottom, tracking data is rendered for the nucleus shown above. Inset shows example of calculated 3D separation distance between the two labels. Scale bar is 1 μm. (**B**) Normalized histogram of 3D separation distance for *Sox2*-SCR ESCs demonstrates a single peak (Hartigan's Dip Test for multimodality, p=1). Schematic for an hypothetical looping enhancer-promoter pair is shown as an inset, with two peaks. Tan box indicates regime where distance measurement error is expected to be greater than 50%. (**C**) Cumulative density of 3D separation distance for *Sox2*-SCR versus control comparisons. Mean distance for each sample shown on bottom right. (**D**) Mean 3D separation distance per cell for each label pair. Population means and standard deviations are shown for each sample. Mann-Whitney, *p<0.05, **p<0.01, ***p<0.001.

DOI: https://doi.org/10.7554/eLife.41769.007

*Figure 2 continued on next page*

*Figure 2 continued*

The following figure supplements are available for figure 2:

**Figure supplement 1.** Tracking Lengths for tetO and cuO Spots Across Cell Lines.

DOI: https://doi.org/10.7554/eLife.41769.008

**Figure supplement 2.** Estimate of Localization Precision for cuO and tetO.

DOI: https://doi.org/10.7554/eLife.41769.009

**Figure supplement 3.** Impact of Localization Precision on 3D Distance Measurements.

DOI: https://doi.org/10.7554/eLife.41769.010

localization precision supported precision of at least this great. These precision estimates translate to an uncertainty in measured 3D distance between cuO/CymR and tetO/TetR of between 40–50 nm (*Figure 2—figure supplement 3*). This localization uncertainty degrades the accuracy of very small distance measurements; distances below 55 nm are dominated by the noise component (i.e. >50% error, *Figure 2—figure supplement 3*). Thus, our experiments are likely to inaccurately describe the 3D separate distance of structures below this value.

Importantly, the cuO and tetO labels are located kilobases away from the *Sox2* promoter and SCR. Hence, these labels imperfectly report on the true locations of the *Sox2* promoter and SCR and may be influenced by other confounding factors, such as the degree of local chromatin compaction. Other potential sources of error include position blurring caused by locus movement during the 30 ms exposure and possible non-diffraction limited behavior of the cuO/tetO arrays. Due to these factors, we expect greater uncertainty regarding how measured distances between cuO/tetO translate to the underlying positions of *Sox2*/SCR than is predicted solely by our localization precision3C data demonstrate enriched contacts between *Sox2* and SCR (*Beagan et al., 2017*; *Bonev et al., 2017*; *de Wit et al., 2015*; *Kieffer-Kwon et al., 2013*; *Mumbach et al., 2016*; *Phillips-Cremins et al., 2013*; *Zhou et al., 2014*), supporting the possibility of a looped locus configuration with *Sox2* and SCR juxtaposed in 3D space. A mixture of looped and unlooped configurations across the population might be expected to produce a multimodal distance distribution with short and large distance peaks representing looped and unlooped states, respectively, as was recently observed for an enhancer system in *Drosophila* (*Chen et al., 2018*). We visualized the measured distances between cuO and tetO in the *Sox2*-SCR configuration as a histogram. This analysis revealed a unimodal distribution with positive skew (Hartigan's Dip Test for multimodality, p=1). On average, *Sox2*/SCR labels are separated by a few hundred nanometers in the ESC nucleus (mean = 339 nm, *Figure 2B*). Infrequently, we observed the *Sox2* region adopt an extended conformation, leading to considerable *Sox2*/SCR separation distance (2.1% of measurements > 750 nm, 0.35% of measurements > 1 µm).

One possible interpretation of a unimodal distance distribution is that the *Sox2*/SCR pair exists predominantly in an interacting state. To investigate this possibility, we repeated this analysis with our two control locus pairs. We found that, while one control pair (Control-Control) did show increased separation distance as compared to *Sox2*/SCR, our other control set (SCR-Control), consisting of the SCR paired with a non-specific partner, showed a similar distribution to *Sox2*/SCR (*Figure 2C*). Indeed, no significant differences between *Sox2*-SCR and SCR –Control were found when comparing the mean distance per cell, while Control-Control demonstrated significantly increased distances (*Figure 2D*). Reinspection of chromosomal contact maps revealed evidence for a topological boundary, potentially established by the SCR element, separating the two labeled regions in the Control-Control configuration (*Figure 1A*), which could account for the elevated 3D distances measured for Control-Control, as has been observed for genomic loci separated by TAD boundaries (*Dixon et al., 2012*; *Nora et al., 2012*). These results suggest that SCR does not show greater proximity to the *Sox2* gene than to a non-specific control.

To further exclude the possibility that our measurements reflected a constitutive interaction state, we sought to estimate the distance profile for a static *Sox2*/SCR interaction. To this end, we used CRISPR/Cas9 to delete a ~ 111 kb fragment between the cuO and tetO labels in the *Sox2*-SCR configuration, leaving a 14 kb tether between the labels (*Figure 1—figure supplement 1*). This is similar in length to the effective tether (~17 kb) between labels expected during a direct interaction between the *Sox2* TSS and the center of the SCR. Visualization of this label configuration in living ESCs demonstrated a significant shift to more proximal distance values (*Figure 2C,D*). These results are consistent with our expectation that a direct *Sox2*/SCR interaction would be confined shorter 3D

distances than those observed for the *Sox2*-SCR pair and validate our experimental capacity to measure these differences. Taken together, these data demonstrate no unique spatial characteristics for the *Sox2*-SCR pair in ESCs. While these observations could suggest very infrequent interaction events, they also may allude to fundamental differences between spatial proximity and the features captured by proximity ligation using 3C approaches (see DISCUSSION).

## Differentiation of ESCs to diverse lineages correlates with *Sox2* locus compaction

We next differentiated our modified cell lines in order to determine how *Sox2* locus organization is altered upon cellular differentiation (*Figure 3A*). To this end, we derived neural precursor cells (NPCs), a cell-type that maintains *Sox2* expression despite inactivation of the SCR and reduced *Sox2*/SCR contacts by chromosome conformation capture carbon copy (5C) (*Figure 3B*) (*Beagan et al., 2017*). We validated that our NPC lines expressed NPC marker genes and demonstrated their ability to differentiate into both neurons and astrocytes (*Figure 3—figure supplement 1*). As an additional comparison, we differentiated our ESC lines into FLK1$^+$/PDGFR$\alpha^+$ mesodermal precursors (MES), a cell type which downregulates *Sox2* expression and inactivates the SCR element (*Figure 3B*). Interestingly, we observed that all label pairs embedded in the *Sox2* locus showed greater proximity in differentiated cells compared to ESCs (*Figure 3C*). These changes were significant when comparing mean distances per cell between label pairs in NPCs or MES with ESCs (*Figure 3D*). These data suggest the entire *Sox2* locus adopts a more compact conformation upon ESC differentiation, regardless of transcriptional status of *Sox2*.

To explore if compaction of the *Sox2* locus conformation might be driven by inactivation of the SCR element (which occurs in both NPCs and MES) or could be driven by other factors related to cellular differentiation, we generated a heterozygous genetic deletion of the SCR element on the 129 allele in ESCs using CRISPR/Cas9 (*Figure 1—figure supplement 1*, *Figure 3—figure supplement 2*). These cells show no signs of differentiation and maintained naive ESC morphology, consistent with previous studies (*Zhou et al., 2014*). Moreover, SCR deletion led to reduction of *Sox2* expression from the *cis* allele to undetectable levels by qPCR (*Figure 1—figure supplement 2*). Live-cell visualization of the cuO and tetO labels in these cells demonstrated a slight shift in 3D distances towards greater proximity; however, this shift was small compared to that seen after differentiation to NPCs or MES (*Figure 3—figure supplement 2*). Hierarchical clustering analysis of the similarity between distance histograms revealed that SCR-deleted ESCs were most similar to other ESC lines (*Figure 3—figure supplement 2*). These observations suggest that *Sox2* locus organization is significantly altered with ESC differentiation but largely robust to changes in *Sox2* or SCR activity.

## Slow *Sox2* locus conformation dynamics lead to limited exploration and variable enhancer encounters

We next investigated the dynamics of *Sox2* spatial organization and focused our analysis of the ESC state. While all three label pairs showed comparable distance profiles across the cell population, we observed striking variation in locus organization between individual cells (*Figure 4A,B*, *Video 2*). We observed label pairs in prolonged compact or extended conformations as well as gradual or sharp transitions between the two (*Figure 4A*). However, few label pairs explored their entire range – the distance spread observed across our cell population – during our imaging window (~25 min), demonstrating that *Sox2* locus conformation dynamics are slow over tens of minutes.

To better understand this phenomenon, we investigated the dynamic properties of our *Sox2*-SCR label pair, as well as both control pairs. Both relative step sizes (defined as the 3D displacement of the cuO label between frames if the tetO location is fixed) and the change in 3D separation distance between frames were significant (e.g.180 nm and 79 nm, respectively, for the *Sox2*-SCR pair, 20 s per frame, *Figure 4—figure supplement 1*). We also computed the autocorrelation function. The autocorrelation function describes the correlation between measurements separated by various lag times and can be utilized to quantify memory or inertia in single cell quantities (e.g. protein levels) compared to the population average (*Sigal et al., 2006*) (*Figure 4C*). Autocorrelation values near one are expected between closely spaced measurements, decaying towards zero for larger lag times. An autocorrelation coefficient of zero indicates that the underlying process has randomized

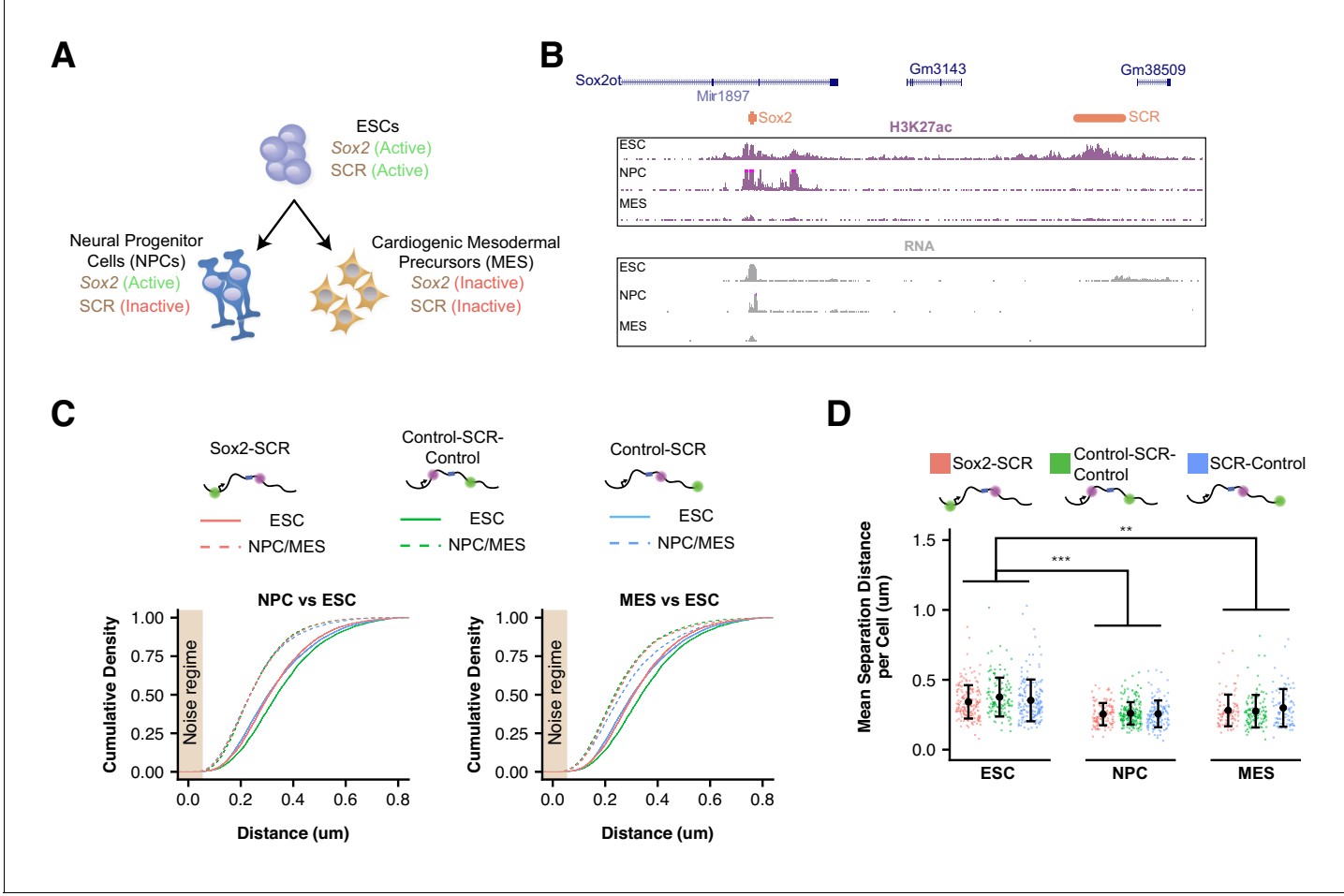

**Figure 3.** *Sox2* Locus Compacts upon ESC Differentiation. (**A**) ESCs were differentiated into neural progenitor cells (NPCs), which maintain expression of *Sox2* but inactivate the SCR, and cardiogenic mesodermal precursors (MES), which inactivate both *Sox2* and the SCR. (**B**) Browser tracks of H3K27ac and RNA-seq data from ESCs, NPCs, and MES demonstrate the activation status of *Sox2* and SCR in each cell type. Y-axis is 0–5 reads per million for H3K27ac data and 0–10 reads per million for RNA-seq data. (**C**) Cumulative density of 3D separation distance for *Sox2*-SCR and two control pairs for NPCs (left) and MES (right). ESC data are shown for comparison as solid lines on each graph and reproduced from ***Figure 2C***. Tan box indicates regime where distance measurement error is expected to be greater than 50%. (**D**) Mean 3D separation distance per cell for each label pair, organized by cell type. Statistical analysis is for each matched pair-wise comparison between cell types. All p-values are below reported value. Mann-Whitney (**p<0.01, ***p<0.001). H3K27ac data from GSE47949 (***Wamstad et al., 2012***) and GSE24164 (***Creyghton et al., 2010***). RNAseq data from GSE47949 and GSE44067 (***Zhang et al., 2013***).

DOI: https://doi.org/10.7554/eLife.41769.011

The following figure supplements are available for figure 3:

**Figure supplement 1.** Characterization of ESC-derived Neural Progenitor Cell Lines.
DOI: https://doi.org/10.7554/eLife.41769.012

**Figure supplement 2.** SCR Inactivation Does Not Drive Locus Compaction Upon Differentiation.
DOI: https://doi.org/10.7554/eLife.41769.013

during the time lag between the relevant measurements. Computation of the autocorrelation function for each label pair revealed a monotonic decay with increasing lag times (***Figure 4D***). We observe an initial rapid reduction in autocorrelation in the small time lag regime, driven by a period of effective local exploration. As our probes begin to oversample the local environment (1–2 mins), the autocorrelation decay slows, reflecting the constraint on locus diffusion within the nuclear environment. Interestingly, at long time lags (>10 mins), the autocorrelation function for both control pairs appears to flatten to a slope of zero, suggesting that conformational memory for some loci may be quite long-lived. These data suggest oversampling of the local environment by individual

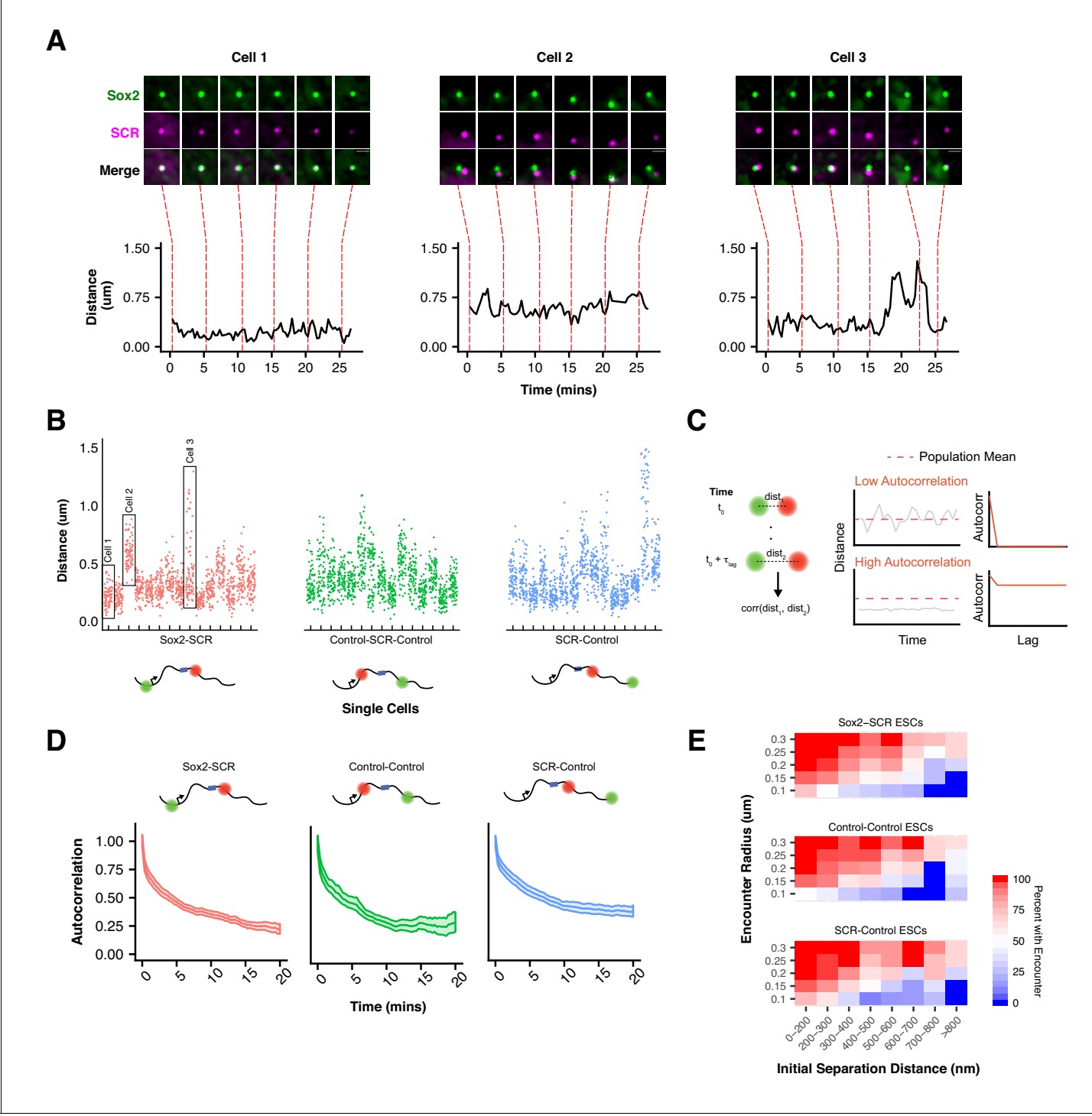

**Figure 4.** Slow *Sox2* Locus Conformation Dynamics Lead to Limited Exploration and Variable Encounters. (**A**) Maximum-intensity projection images (top) centered on the *Sox2* locus and associated 3D distance measurements (bottom) highlight distinct conformations and dynamics of the *Sox2* locus across cells. Scale bar is 1 μm. (**B**) 3D separation distance measurements for individual cells for *Sox2*-SCR, Control-Control, and SCR-Control highlight the heterogeneity of *Sox2* locus organization across the cell population. The three cells depicted in A are boxed. (**C**) Cartoon description of autocorrelation analysis. Distance measurement between two time points are correlated using population statistics, revealing the time scale over which local measurements diverge from the population mean. A cell with low autocorrelation will randomly fluctuate around the population mean, leading the autocorrelation function to quickly decay to zero. A cell with high autocorrelation will deviate substantially from the expected value, only slowly relaxing back to the population mean. In this case, the autocorrelation function will stay significantly above zero for large lag times. (**D**) Autocorrelation function for *Sox2*-SCR, Control-Control, and SCR-Control pairs demonstrates significant autocorrelation at large lag times, indicating significant memory in 3D

*Figure 4 continued on next page*

*Figure 4 continued*

conformation across a 20 min window. The plotted values are mean ± 95% CI. **E**) Percent of cells with an encounter between tetO and cuO labels shown as a function of the initial separation distance measured for the cell. Likelihood of an encounter depends on the initial conformation of the locus across all label pairs and encounter thresholds.

DOI: https://doi.org/10.7554/eLife.41769.014

The following figure supplement is available for figure 4:

**Figure supplement 1.** Dynamics Statistics for Each Sox2 Locus Pair in ESCs.

DOI: https://doi.org/10.7554/eLife.41769.015

loci within the *Sox2* region and are consistent with current physical models of chromatin (*Dekker and Mirny, 2016*) and the viscoelastic nature of the nucleoplasm (*Lucas et al., 2014*).

An important implication of this behavior of chromatin is that encounters between loci are highly dependent on the initial configuration of the genomic region (*Figure 4E*). This can be seen by investigating the proportion of cells where the cuO and tetO labels have at least one encounter (defined by a separation distance below a proximity threshold). For instance, while 73% of *Sox2*-SCR pairs that start within 200 nm of each other are observed to have at least one encounter below 100 nm over the 25 min imaging window, this drops to 18% for pairs that start greater than 600 nm away. This trend is observed across label pairs and is robust to threshold value (*Figure 4E*). Such behavior could have important consequences for gene regulation by enhancer-promoter interactions. Given the observed inertia in locus conformation, enhancer proximity, and therefore the capacity for direct enhancer-promoter contact, is likely to be highly variable across time within a cell and between cells within a population.

## Visualization of *Sox2* transcriptional bursts in living ESCs

We next explored the temporal relation between 3D organization of the *Sox2* locus and transcription. To this end, we utilized the well-established MS2 reporter system to directly visualize nascent transcription in single living ESCs (*Bertrand et al., 1998*). Using CRISPR/Cas9 genome engineering, we replaced the endogenous 129 *Sox2* allele with a modified version that includes a P2A-puromycin resistance gene fusion and 24 MS2 stem loops inserted into the 3' UTR of the *Sox2* gene (*Figure 5— figure supplement 1*). We generated this MS2 reporter allele in our *Sox2*-SCR labeled cell line to generate *Sox2*-8C$^{cuO/+}$, *Sox2*-117T$^{tetO/+}$, *Sox2*$^{MS2/WT}$ ESCs (or simply *Sox2*-MS2 ESCs). Transcription levels derived from the *Sox2*-MS2 reporter allele were 35% of those from the untargeted 129 allele (*Figure 1—figure supplement 2*), potentially due to reduced stability of transcripts labeled with MS2 stem loops (*Ochiai et al., 2014*). Western blotting of Sox2-MS2 lysate revealed a SOX2 doublet as expected, suggesting proper expression of both wild-type SOX2 and the SOX-P2A fusion (*Figure 5—figure supplement 1*).

We first characterized the transcriptional activity of *Sox2*-MS2 reporter allele. We co-expressed a tandem-dimer of the MS2 coat protein fused with 2 copies of tagRFP-T (tdMS2cp-tagRFP-Tx2), TetR fused with 2 copies of GFP (TetR-GFPx2), and CymR fused with 2 copies of Halo tag (CymR-Halox2) in *Sox2*-MS2 ESCs. These ESCs enabled simultaneous visualization of the labels adjacent to the *Sox2* promoter and SCR, as well as nascent *Sox2* transcription in living ESCs when imaged in the presence of the Halo-tag ligand JF646 (*Grimm et al., 2015*) (*Figure 5A*). Time-lapse confocal microscopy revealed bright flashes of MS2cp signal in the ESC nucleus, which occurred in spatial proximity to the cuO and tetO labels, and were similar to the MS2 transcriptional bursts observed elsewhere (*Bothma et al., 2014*; *Chubb et al.,*

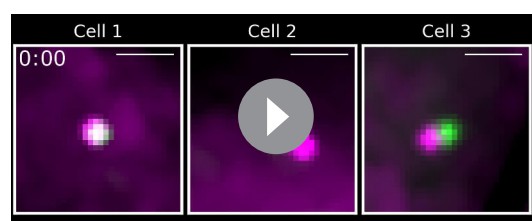

**Video 2.** Variability in *Sox2* Locus Organization Across Cells. Maximum-intensity Z projection of 3D confocal Z-stacks of cuO/CymR (green) and tetO/TetR (magenta) labeling the *Sox2* promoter region and SCR, respectively for three individual cells highlighted in *Figure 3*. The distance range explored by Cell1 and Cell2 is limited, while Cell3 shows large, abrupt changes in distance. Scale bar is 1 μm.

DOI: https://doi.org/10.7554/eLife.41769.016

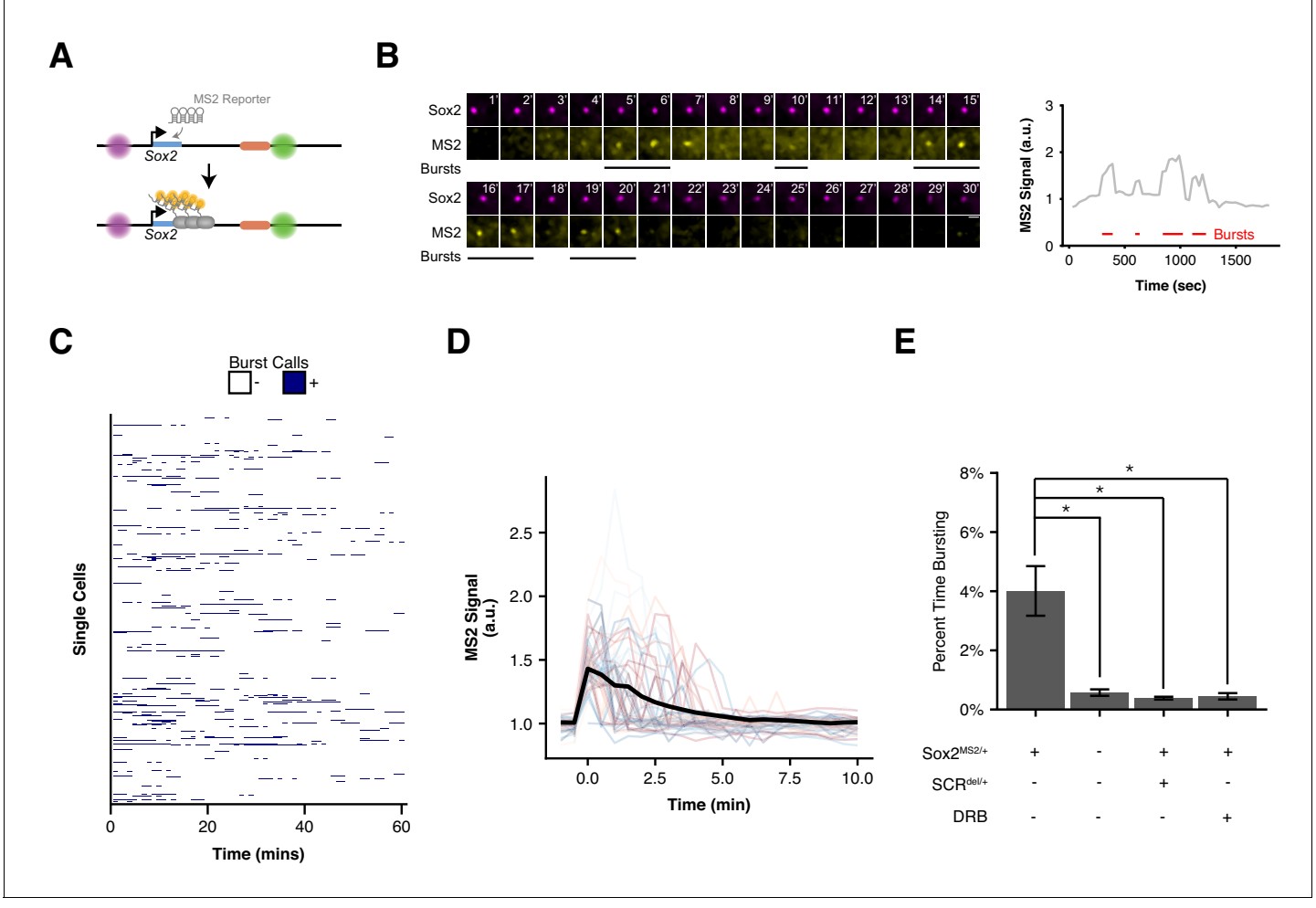

**Figure 5.** Visualizing *Sox2* Expression in Single Living ESCs Reveals Intermittent Bursts of Transcription. (**A**) *Sox2* locus with cuO-labeled *Sox2* promoter and tetO-labeled SCR was further modified to introduce an MS2 transcriptional reporter cassette into the *Sox2* gene. Transcription of *Sox2* leads to visible spot at the *Sox2* gene due to binding and clustering of MS2 coat protein to the MS2 hairpin sequence. (**B**) Maximum-intensity projection images centered on the *Sox2* promoter (cuO) show intermittent bursts of MS2 signal, which are quantified on the right. Scale bar is 1 µm. (**C**) Single cell trajectories of *Sox2* transcriptional bursts as representatively shown in B. (**D**) Aligned *Sox2* transcriptional bursts. Randomly selected *Sox2* bursts are shown as color traces (n = 50). Black line is mean MS2 signal for all annotated bursts. (**E**) Percent time *Sox2* transcriptional bursting for various experimental conditions. Bars are mean ± standard error of ≥3 independent experiments. $Sox2^{MS2/+}$ indicates cell line harbors the *Sox2*-MS2 reporter allele. $SCR^{del/+}$ indicates presence of an SCR deletion in cis with the *Sox2*-MS2 reporter. DRB indicates treatment with the transcriptional inhibitor 5,6-Dichloro-1-β-D-ribofuranosylbenzimidazole (DRB).

DOI: https://doi.org/10.7554/eLife.41769.017

The following figure supplement is available for figure 5:

**Figure supplement 1.** Generation and Characterization of Sox2-MS2 Transcriptional Reporter ESCs.

DOI: https://doi.org/10.7554/eLife.41769.018

*2006*; *Lionnet et al., 2011*; *Martin et al., 2013*; *Ochiai et al., 2014*). These results suggested the *Sox2* MS2 reporter allele enables visualization of *Sox2* transcription.

Using our pipeline, we identified a total of 603 individual bursts across 1,208 cells (*Figure 5B*, *Supplementary files 5,6*, *Video 3*). We found *Sox2* transcriptional activity to be sporadic both between cells and within individual cells across time (*Figure 5C*). Nearly two-thirds (66.1%) of nuclei lacked detectable *Sox2* transcription during our 30 min imaging window, with the majority of remaining cells demonstrating transcriptional activity in less than 20% of frames (29.3%, *Figure 5—figure supplement 1*). However, we did observe rare cells that demonstrated robust transcriptional activity in greater than half the observed frames (0.25% of cells, *Video 4*). We also found substantial variability in the intensity of transcriptional bursts and their duration (*Figure 5D*). As a population,

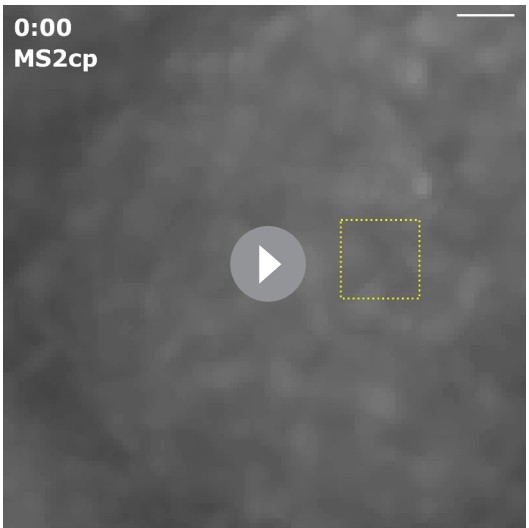

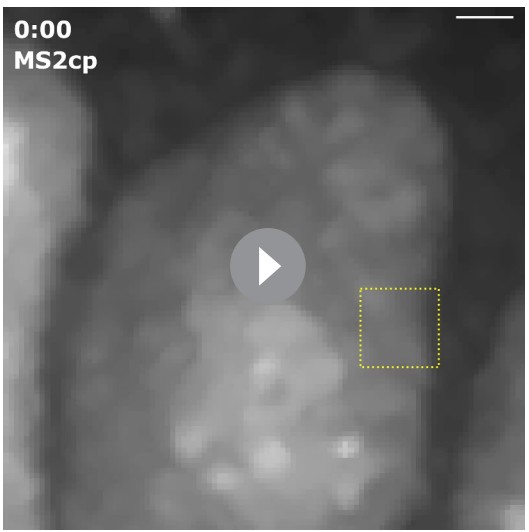

**Video 3.** Identification of *Sox2* Transcriptional Bursts in mESCs. Maximum-intensity Z projection of 3D confocal Z-stacks of a tandem dimer of MS2 coat protein fused with two copies of tagRFP-T. The dashed yellow box highlights the ROI used for burst detection in our automated analysis pipeline, centered on the location of the Sox2 promoter (cuO/CymR location, not shown). Detected bursts are highlighted by red circles centered on the burst location, with color intensity indicating burst intensity. Scale bar is 1 μm.
DOI: https://doi.org/10.7554/eLife.41769.019

**Video 4.** High Transcriptional Output from *Sox2* Locus. Maximum-intensity Z projection of 3D confocal Z-stacks of a tandem dimer of MS2 coat protein fused with two copies of tagRFP-T demonstrate a period of high transcriptional activity for the highlighted *Sox2* gene. The dashed yellow box highlights the ROI used for burst detection in our automated analysis pipeline, centered on the location of the *Sox2* promoter (cuO/CymR location, not shown). Detected bursts are highlighted by red circles centered on the burst location, with color intensity indicating burst intensity. Scale bar is 1 μm.
DOI: https://doi.org/10.7554/eLife.41769.020

we found Sox2-MS2 ESCs spent 4% of their time with a detectable MS2 burst (*Figure 5E*). Thus, our live-cell measurements of *Sox2* transcription suggest short, intermittent transcriptional activity in ESCs.

To ensure that our MS2 analysis identified bona fide transcriptional activity, we repeated our analysis in a number of control contexts. First, we measured bursting frequency in ESCs that expressed the MS2 coat protein but lacked the *Sox2*-MS2 reporter allele($Sox2$-8C$^{cuO/+}$, $Sox2$-117T$^{tetO/+}$, $Sox2^{WT/WT}$). Second, we measured bursting frequency in *Sox2*-MS2 ESCs that harbored an SCR deletion in cis ($Sox2$-8C$^{cuO/+}$, $Sox2$-117T$^{tetO/+}$, $Sox2^{MS/WT}$, SCR$^{del/+}$). Third, we measured bursting frequency in *Sox2*-MS2 ESCs that were treated with the transcriptional inhibitor 5,6-Dichloro-1-β-D-ribofuranosylbenzimidazole (DRB). In each case, we observed a significant drop in *Sox2* burst frequency (*Figure 5E*). Taken together, these data demonstrate our ability to accurately identify *Sox2* transcriptional events using our MS2 reporter cell line.

### *Sox2* transcription is not associated with SCR proximity

Assuming SCR regulates *Sox2* transcription via the conventional enhancer looping model, we would expect *Sox2* transcriptional activity to occur during interactions or periods of *Sox2*/SCR proximity (*Figure 6A*), given that *Sox2* depends of SCR for its ESC expression. To investigate this prediction, we restricted our analysis to nuclei with single, diffraction-limited spots for the cuO and tetO labels in our *Sox2*-MS2 ESC dataset. We calculated 3D distances between the cuO/tetO and compared single cell distance traces with matched MS2 signal traces. We identified some transcriptionally active cells that showed prolonged proximity of the *Sox2*/SCR labels. However, we also observed cells which showed robust transcriptional bursting despite a prolonged extended conformation of the *Sox2* region, driving *Sox2*/SCR distance above the population average for the duration of our 30 min imaging window (*Figure 6B*, *Video 5*). We binned time points according to the measured distance between *Sox2* and SCR and calculated the percent time spent bursting for each bin and found

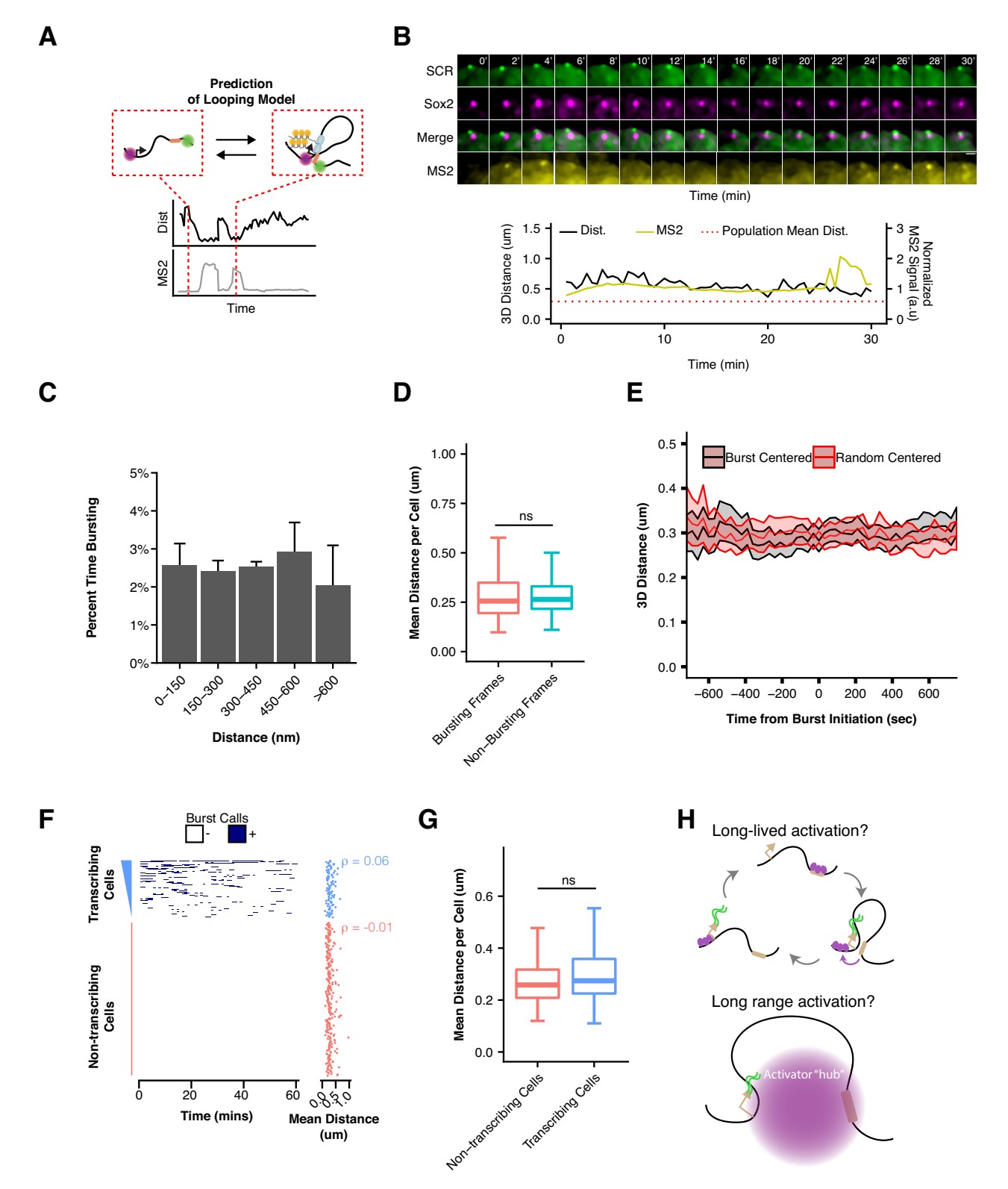

**Figure 6.** *Sox2* Transcription Is Not Associated with SCR Proximity. (**A**) Schematic illustrating the expected relation between *Sox2*/SCR distance and MS2 transcription for a looping enhancer model. (**B**) Maximum-intensity projection images centered on the *Sox2* promoter (cuO) show transcriptional activity without correlation to *Sox2*/SCR distance changes. The measured distance and MS2 signal are shown at bottom. The mean separation distance across the cell population is shown as a dotted red line. Scale bar is 1 μm. (**C**) Percent time with *Sox2* transcriptional burst as a function of *Sox2*/SCR

*Figure 6 continued*

distance. Weighted mean + SE for seven experiments are shown. Weights were determined based on the proportion of frames in each bin contributed by individual experiments. (**D**) Mean separation distance per cell, separated into bursting and non-bursting frames. (Mann-Whitney, p=0.68). (**E**) Mean separation distance across a 25 min window for all transcriptional bursts (black) or randomly select time points (red), aligned according the burst initiation frame. Values plotted are mean ± 95% CI. (**F**) Single cell trajectories of *Sox2* transcriptional bursts ranked by number of bursting frames per cell. At right, matched mean separation distances for each cell shown at left. Spearman's correlation coefficient for each is shown. (**G**) Mean separation distance per cell for transcribing and non-transcribing cells. (Mann-Whitney, p=0.15). (**H**) Potential models of SCR regulation of *Sox2* that would uncouple *Sox2*/SCR proximity from transcriptional activity. Above, SCR leads to long-lived activation of the *Sox2* promoter that can persist long after *Sox2*/SCR contact is disassembled. Below, SCR nucleates a large hub of activator proteins that can modify the *Sox2* promoter environment despite large distances between *Sox2* and SCR.

DOI: https://doi.org/10.7554/eLife.41769.021

The following figure supplement is available for figure 6:

**Figure supplement 1.** Relative Displacement between Frames for Bursting and Non-Burst Time Points.

DOI: https://doi.org/10.7554/eLife.41769.022

that all bins showed similar transcriptional activity (*Figure 6C*). Furthermore, segregating time points into bursting and non-bursting frames for each cell demonstrated no significant differences between the two groups (*Figure 6D*, Mann-Whitney, p=0.68).

We next considered the possibility that *Sox2*/SCR proximity might precede transcriptional bursting by a characteristic time. This might be expected if there are characteristic delays for transcription complex assembly or to allow for elongation to the 3' MS2 sequence (based on an estimated elongation rate of 30–100 nt/sec [*Fuchs et al., 2014*], it would require ~ 0.5–2 min for polymerase to reach the 3' end of the MS2 array). We identified the initiation point for all bursts in our dataset and considered a 25 min window centered at each burst initiation event. Alignment and meta-analysis of these bursts showed little change in *Sox2*/SCR distance across the time window. To determine if *Sox2*/SCR distance significantly deviated from expected values across transcriptional bursts, we compared aligned bursts to a randomly shuffled control dataset and found no significant differences between the burst-centered and random-centered analysis (*Figure 6E*, *Supplementary file 7*). This analysis suggests *Sox2*/SCR proximity and *Sox2* transcription is not separated by a characteristic lag within the time frame considered.

Finally, given the high degree of cell-to-cell variability in *Sox2* locus organization, we investigated whether cells with greater average *Sox2*-SCR proximity, which would enable more frequent *Sox2*/SCR encounters, demonstrated higher transcriptional activity. We rank ordered cells based on cumulative transcriptional activity (i.e. number of transcriptionally active frames) and compared mean *Sox2*/SCR distance per cell (*Figure 6F*). As expected, non-transcribing cells showed no correlation between order and distance, given the ordering within this group was essentially random (Spearman's ρ = −0.01). However, transcribing cells also showed no correlation between transcriptional activity and distance (Spearman's ρ = 0.06). As a group, transcribing cells demonstrated no significant difference in mean *Sox2*/SCR separation distance compared to non-transcribing cells (*Figure 6G*, Mann-Whitney, p=0.15). These data suggest little relation between the 3D conformation of *Sox2* relative to the SCR enhancer and its transcriptional output. Thus, our data indicate SCR is unlikely to directly activate *Sox2* transcription through contact with its promoter.

## Discussion

We have investigated the dynamic 3D organization and underlying transcriptional activity of the

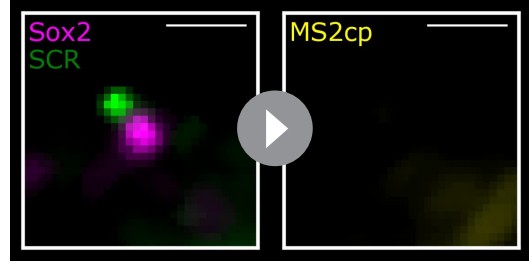

**Video 5.** *Sox2* Transcriptional Bursts in the Absence of SCR Proximity. Maximum-intensity Z projection of 3D confocal Z-stacks of cuO/CymR (green) and tetO/TetR (magenta) labeling the *Sox2* promoter region and SCR, respectively (left), and MS2 coat protein highlighting *Sox2* transcriptional activity (right). We detect clear *Sox2* transcriptional bursts despite no colocalization of the Sox2/SCR labels. Scale bar is 1 μm.

DOI: https://doi.org/10.7554/eLife.41769.023

established enhancer-gene pair *Sox2* and SCR. Interestingly, we observe few unique spatial characteristics for Sox2/SCR in ESCs; observed distance distributions and their spatial dynamics for SCR and the *Sox2* promoter region are similar to those observed between SCR and an equally-spaced non-specific region. In contrast, 3C-based assays have identified enriched contacts between Sox2/SCR as compared to the surrounding neighborhood. We note that these results need not be incompatible. Proximity ligation (3C) and separation distance (microscopy) are distinct measures of chromatin structure with unique biases, assumptions, and limitations, and thus provide snapshots of chromatin architecture that may differ (*Dekker, 2016*; *Fudenberg and Imakaev, 2017*; *Giorgetti and Heard, 2016*). 3C-based assays often utilize millions of cells and so may capture rare conformations in the cell population; these rare conformations would have minimal impact on overall distance distributions constructed using microscopy. Moreover, it remains unclear what spatial proximity is required to enable ligation events during 3C, and this property may differ for distinct genomic regions. Indeed, enrichment of *Sox2*/SCR contacts in 3C assays may reflect only subtle differences in very proximal conformations (e.g. < 50 nm), conformations unlikely to be accurately represented by our microscopy measurements due to technical limitations in localization precision and uncertainty. Alternatively, large macromolecular bridges or hubs may enable crosslinking and ligation over larger distances that need not demonstrate pronounced spatial proximity, as recently demonstrated (*Quinodoz et al., 2018*). Moreover, chromatin composition and accessibility are likely to influence key features for 3C and microscopy experiments, such as crosslinkability, distances permissive for proximity ligation, and the scaling of spatial distances with genomic distance. All of these sources of uncertainty raise questions regarding how features from 3C and microscopy translate between assays and to the underlying chromatin structure. While a comprehensive picture of *Sox2* locus organization remains out of view, our study provides guidance as to what structures are unlikely. For instance, the absence of enhanced proximity between the *Sox2* and SCR pair suggests a prolonged, proximal conformation established by stable, direct pairing of the *Sox2* promoter with SCR is unlikely to be the predominant structure in ESCs.

Surprisingly, we also observe no association between *Sox2*/SCR proximity and *Sox2* transcription in real time. Indeed, we detect no correlation between transcriptional activity and instantaneous *Sox2*/SCR distances, no reduction in *Sox2*/SCR distances prior to transcriptional bursts, and no tendency for transcriptionally active cells to display reduced *Sox2*/SCR distance. It is important to note that we cannot exclude the importance of direct Sox2/SCR contacts in Sox2 activation. If these events lead to a complex, multi-step activation process with stochastic delays between steps, it is plausible that enhancer-promoter contact and transcriptional output could be temporally decoupled and demonstrate the poor correlation between *Sox2*/SCR proximity and transcriptional activity that we observe. Furthermore, SCR contacts could be important for long-lived activation of the Sox2 promoter, which could persist after disassembly of these interactions (*Figure 6H*, top). This mechanism might be achieved through delivery of durable factors (e.g. chromatin modifiers) to the *Sox2* promoter during contact, and might explain why disruption of DNA loops genome-wide through acute RAD21 degradation leads to only modest changes in nascent transcription after 6 hr (*Rao et al., 2017*).

The *Sox2* locus displays distinct behavior from an enhancer reporter recently used to explore the regulatory logic of the even-skipped (*eve*) enhancers in *Drosophila* embryos. In this study, the authors integrated an enhancer reporter ~ 142 kb upstream of *eve* locus and promoted pairing between the two loci by including an ectopic insulator sequence, which pairs with a similar sequence embedded near the *eve* enhancers. In this system, the authors observe both bimodality in distance measurements as well as clear correlation between enhancer-reporter proximity and reporter transcription. While it is not yet clear why these systems behave so differently, we note the considerable differences in the 3D distances we report for *Sox2* (339 nm for *Sox2*/SCR) and those reported for the even-skipped reporter (709 nm for unpaired and 353 nm for paired). It seems plausible that the more extended conformation of the *Drosophila* chromosome necessitates pairing in order to bring the *eve* enhancer sufficiently close the reporter, particularly for enhancers evolved to function within 10 kb of their target gene. Our analysis suggests that most *Sox2*/SCR loci reside within this distance range, perhaps lowering the importance of locus conformation for SCR function. Indeed, SCR transcriptional control does demonstrate proximity dependence on some scale, as SCR ablation is not compensated for by a normal copy located on the homologous chromosome (*Li et al., 2014*; *Zhou et al., 2014*). In other contexts, such as during olfactory receptor gene choice or transvection

in *Drosophila*, regulation can occur over very large distances in cis (~80 Mb) or in trans, and transcriptional activity may be more closely tied to pairing events that promote spatial proximity, as recently demonstrated for the latter (*Horta et al., 2018*; *Lim et al., 2018*; *Markenscoff-Papadimitriou et al., 2014*). Hence, genomic interactions and other features of genome topology may differ in importance depending of the spatial distances navigated by enhancer-gene pairs.

Our observations also open the possibility that direct contacts between *Sox2* and SCR are dispensable for SCR function. Numerous mechanisms for long-range communication between enhancers and promoters have been proposed (*Bulger and Groudine, 2010*). For example, SCR may play a critical role in the nucleation and spreading of important epigenetic activators and chromatin accessibility, establishing a permissive environment of *Sox2* transcription. An intriguing mechanism for action at a distance comes from recent observations that super-enhancers are capable of nucleating large (>300 nm), phase-separated condensates of coactivators, chromatin regulators, and transcription complexes (*Cho et al., 2018*; *Sabari et al., 2018*). SCR is a bona fide super-enhancer in ESCs (*Whyte et al., 2013*). Thus, SCR may deliver activation factors over hundreds of nanometers through inclusion of the *Sox2* promoter into an activator hub or condensate (*Figure 6H*, bottom). Such a mechanism would present a number of challenges for achieving precise transcriptional control, most notably how SCR selectivity for *Sox2* activation is achieved. Nevertheless, future studies that couple visualization of the *Sox2* locus with that of important molecular components of transcriptional activation are likely to be essential in decoding how the SCR element achieves tight expression control of this essential pluripotency gene.

# Materials and methods

**Key resources table**

| Reagent type (species) or resource | Designation | Source or reference | Identifiers | Additional information |
|---|---|---|---|---|
| Cell line (*M. musculus*) | 129/Cast F1 ESCs | PMID: 9298902 | | |
| Cell line (*M. musculus*) | E14 ESCs | PMID: 3821905 | RRID:CVCL_C320 | |
| Cell line (*M. musculus*) | Sox2-SCR ESCs | this paper | | 129/Cast F1 ESCs with cuO array inserted 8 kb centromeric to *Sox2* TSS and tetO array inserted 117 kb telomeric to *Sox2* TSS on the 129 allele |
| Cell line (*M. musculus*) | Sox2-SCR ESCs; CymR-GFP; TetR-tdTom | this paper | | 129/Cast F1 ESCs with cuO array inserted 8 kb centromeric to *Sox2* TSS and tetO array inserted 117 kb telomeric to *Sox2* TSS on the 129 allele. Cells stably express ePiggyBac vectors epB-UbC-CymRV5-nls-GFP-DEx2 and epB-CAG-TetRFlag-nls-tdTom-DEx4 |
| Cell line (*M. musculus*) | Control-Control ESCs | this paper | | 129/Cast F1 ESCs with tetO array inserted 43 kb telomeric to *Sox2* TSS and cuO array inserted 164 kb telomeric to *Sox2* TSS on the 129 allele |

*Continued on next page*

*Continued*

| Reagent type (species) or resource | Designation | Source or reference | Identifiers | Additional information |
|---|---|---|---|---|
| Cell line (*M. musculus*) | Control-Control ESCs | this paper | | 129/Cast F1 ESCs with tetO array inserted 43 kb telomeric to *Sox2* TSS and cuO array inserted 164 kb telomeric to *Sox2* TSS on the 129 allele. Cells stably express ePiggyBac vectors epB-UbC-CymRV5-nls-GFP-DEx2 and epB-CAG-TetRFlag-nls-tdTom-DEx4 |
| Cell line (*M. musculus*) | SCR-Control ESCs | this paper | | 129/Cast F1 ESCs with tetO array inserted 117 kb telomeric to *Sox2* TSS and cuO array inserted 242 kb telomeric to *Sox2* TSS on the 129 allele |
| Cell line (*M. musculus*) | SCR-Control ESCs | this paper | | 129/Cast F1 ESCs with tetO array inserted 117 kb telomeric to *Sox2* TSS and cuO array inserted 242 kb telomeric to *Sox2* TSS on the 129 allele. Cells stably express ePiggyBac vectors epB-UbC-CymRV5-nls-GFP-DEx2 and epB-CAG-TetRFlag-nls-td Tom-DEx4 |
| Cell line (*M. musculus*) | SCR deletion ESCs | this paper | | 129/Cast F1 ESCs with cuO array inserted 8 kb centromeric to *Sox2* TSS and tetO array inserted 117 kb telomeric to *Sox2* TSS on the 129 allele. SCR deletion (104 kb-112kb from *Sox2* TSS) is present on 129 allele |
| Cell line (*M. musculus*) | SCR deletion ESCs | this paper | | 129/Cast F1 ESCs with cuO array inserted 8 kb centromeric to *Sox2* TSS and tetO array inserted 117 kb telomeric to *Sox2* TSS on the 129 allele. SCR deletion (104 kb-112kb from *Sox2* TSS) is present on 129 allele. Cells stably express ePiggyBac vectors epB-UbC-CymRV5-nls-GFP-DEx2 and epB-CAG-TetRFlag-nls -tdTom-DEx4 |
| Cell line (*M. musculus*) | Sox2-MS2 ESCs | this paper | | 129/Cast F1 ESCs with cuO array inserted 8 kb centromeric to *Sox2* TSS and tetO array inserted 117 kb telomeric to *Sox2* TSS on the 129 allele. 129 Sox2 allele has been replaced with Sox2-P2A-puro-24xMS2. |

*Continued on next page*

*Continued*

| Reagent type (species) or resource | Designation | Source or reference | Identifiers | Additional information |
|---|---|---|---|---|
| Cell line (*M. musculus*) | Sox2-MS2 ESCs | this paper | | 129/Cast F1 ESCs with cuO array inserted 8 kb centromeric to *Sox2* TSS and tetO array inserted 117 kb telomeric to *Sox2* TSS on the 129 allele. 129 Sox2 allele has been replaced with Sox2-P2A-puro-24xMS2. Cells stably express ePiggyBac vectors epB-UbC-CymRV5-nls-Halox2-DEx4, epB-CAG-TetRFlag-nls-GFPx2, and epB-UbC-tdMS2cp-tagRFP-Tx2 |
| Cell line (*M. musculus*) | Sox2-MS2; SCR deletion ESCs | this paper | | 129/Cast F1 ESCs with cuO array inserted 8 kb centromeric to *Sox2* TSS and tetO array inserted 117 kb telomeric to *Sox2* TSS on the 129 allele. 129 Sox2 allele has been replaced with Sox2-P2A-puro-24xMS2. SCR deletion (104 kb-112kb from Sox2 TSS) is present on 129 allele |
| Cell line (*M. musculus*) | Sox2-MS2; SCR deletion ESCs | this paper | | 129/Cast F1 ESCs with cuO array inserted 8 kb centromeric to *Sox2* TSS and tetO array inserted 117 kb telomeric to *Sox2* TSS on the 129 allele. 129 Sox2 allele has been replaced with Sox2-P2A-puro-24xMS2. SCR deletion (104 kb-112kb from Sox2 TSS) is present on 129 allele. Cells stably express ePiggyBac vectors epB-UbC-CymRV5-nls-Halox2-DEx4, epB-CAG-TetRFlag-nls-GFPx2, and epB-UbC-tdMS2cp-tagRFP-Tx2 |
| Cell line (*M. musculus*) | Sox2-del-SCR ESCs | this paper | | 129/Cast F1 ESCs with cuO array inserted 8 kb centromeric to *Sox2* TSS and tetO array inserted 117 kb telomeric to *Sox2* TSS on the 129 allele. Large deletion (1 kb-112kb from *Sox2* TSS) is present on 129 allele. All genetic distances based on reference genome. |

*Continued on next page*

*Continued*

| Reagent type (species) or resource | Designation | Source or reference | Identifiers | Additional information |
|---|---|---|---|---|
| Cell line (*M. musculus*) | Sox2-del-SCR ESCs | this paper | | 129/Cast F1 ESCs with cuO array inserted 8 kb centromeric to *Sox2* TSS and tetO array inserted 117 kb telomeric to *Sox2* TSS on the 129 allele. Large deletion (1 kb-112kb from *Sox2* TSS) is present on 129 allele. All genetic distances based on reference genome. Cells stably express ePiggyBac vectors epB-UbC-CymRV5 -nls-GFP-DEx2 and epB-CAG -TetRFlag-nls-tdTom-DEx4. |
| Cell line (*M. musculus*) | Sox2-SCR NPCs | this paper | | Neural progenitor cells derived from Sox2-SCR ESCs. Cells stably express ePiggyBac vectors epB-UbC -CymRV5-nls-GFP-DEx2 and epB-CAG-TetRFlag-nls -tdTom-DEx4. |
| Cell line (*M. musculus*) | Sox2-SCR NPCs | this paper | | Neural progenitor cells derived from Sox2-SCR ESCs |
| Cell line (*M. musculus*) | Control-Control NPCs | this paper | | Neural progenitor cells derived from Control-Control ESCs. Cells stably express ePiggyBac vectors epB-UbC-CymRV5-nls-GFP-DEx2 and epB-CAG-TetRFlag- nls-tdTom-DEx4. |
| Cell line (*M. musculus*) | SCR-Control NPCs | this paper | | Neural progenitor cells derived from SCR-Control ESCs |
| Cell line (*M. musculus*) | SCR-Control NPCs | this paper | | Neural progenitor cells derived from SCR-Control ESCs. epB-UbC-CymRV5 -nls-GFP-DEx2 and epB-CAG-TetRFlag-nls-td Tom-DEx4. |
| Antibody | rat monoclonal PE-conjugated anti-PDGFRα | Thermo Fisher | 12-1401-81; RRID:AB_657615 | Flow 1:400 |
| Antibody | mouse monoclonal anti-SOX2 | Santa Cruz | sc-365823; RRID:AB_10842165 | WB 1:1000, IF 1:100 |
| Antibody | rabbit polyclonal anti-PAX6 | Biolegend | 901301; RRID:AB_2565003 | IF 1:100 |
| Antibody | mouse monoclonal anti-TUBB3 | Biolegend | 801201; RRID:AB_2313773 | IF 1:100 |
| Antibody | mouse monoclonal anti-GFAP | Sigma | G3893; RRID:AB_477010 | IF 1:400 |
| Antibody | rabbit polyclonal anti-βactin | Abcam | ab8227; RRID:AB_2305186 | WB 1:2000 |
| Antibody | anti-Flk1 biotin | PMID: 17084363 | | Hybridoma clone D218 Flow 1:100 |

*Continued on next page*

*Continued*

| Reagent type (species) or resource | Designation | Source or reference | Identifiers | Additional information |
|---|---|---|---|---|
| Recombinant DNA reagent | pCAGGS-Bxb1o-nlsFlag | this paper | Addgene: 119901 | Expresses Bxb1 integrase in mammalian cells |
| Recombinant DNA reagent | pDEST-tetOx224_PhiC31attB_loxP-PGKpuro-loxP | this paper | Addgene: 119902 | PhiC31 integration plasmid for tetO array with Neo selection cassette |
| Recombinant DNA reagent | pDEST-cuOx144_Bxb1attB_loxP-PGKpuro-loxP | this paper | Addgene: 119903 | Bxb1 integration plasmid for cuO array with Puro selection cassette |
| Recombinant DNA reagent | pDEST-tetOx224_PhiC31attB_FRT-EF1a-GFP-FRT | this paper | Addgene: 119904 | PhiC31 integration plasmid for tetO array with GFP expression cassette |
| Recombinant DNA reagent | pDEST-cuOx144_Bxb1attB_loxP-EF1a-tagRFP-T-loxP | this paper | Addgene: 119905 | Bxb1 integration plasmid for cuO array with RFP expression cassette |
| Recombinant DNA reagent | epB-UbC_CymRV5-nls-GFP-DEx2 | this paper | Addgene: 119906 | ePiggyBac mammalian expression plasmid for CymR-GFP fusion |
| Recombinant DNA reagent | epB-UbC_CymRV5-nls-Halox2_DEx4 | this paper | Addgene: 119907 | ePiggyBac mammalian expression plasmid for CymR-Halo fusion |
| Recombinant DNA reagent | epB-UbC_tdMS2cp-tagRFP-Tx2 | this paper | Addgene: 119908 | ePiggyBac mammalian expression plasmid for tandem dimer MS2cp-tagRFP-T fusion |
| Recombinant DNA reagent | epB_CAG_TetRFlag-nls-tdTom-DEx4 | this paper | Addgene: 119909 | ePiggyBac mammalian expression plasmid for TetR-tdTom fusion |
| Recombinant DNA reagent | epB_CAG_TetRFlag-nls_GFPx2_DEx2 | this paper | Addgene: 119910 | ePiggyBac mammalian expression plasmid for TetR-GFP fusion |
| Recombinant DNA reagent | ePiggyBac-Transposase | this paper | Addgene: 119911 | Mammalian expression plasmid for the ePiggy Bac transposes |
| Recombinant DNA reagent | pKS_Sox2-P2A-puro-24xMS_targeting_vector_NoPAM | this paper | | Targeting vector for generating Sox2-MS2 allele |
| Recombinant DNA reagent | pX330-Sox2_3′ UTR_gRNA | this paper | | Cas9/sgRNA expression vector with gRNA that targets the *Sox2* 3′ UTR |
| Recombinant DNA reagent | pX330-Sox2-8C_gRNA | this paper | | Cas9/sgRNA expression vector with gRNA that targets 8 kb centromeric to *Sox2* TSS |
| Recombinant DNA reagent | pX330-Sox2-43T_gRNA | this paper | | Cas9/sgRNA expression vector with gRNA that targets 43 kb telomeric to *Sox2* TSS |
| Recombinant DNA reagent | pX330-Sox2-117T_gRNA | this paper | | Cas9/sgRNA expression vector with gRNA that targets 117 kb telomeric to *Sox2* TSS |
| Recombinant DNA reagent | pX330-Sox2-164T_gRNA | this paper | | Cas9/sgRNA expression vector with gRNA that targets 164 kb telomeric to *Sox2* TSS |

*Continued*

| Reagent type (species) or resource | Designation | Source or reference | Identifiers | Additional information |
|---|---|---|---|---|
| Recombinant DNA reagent | pX330-Sox2-104T_gRNA | this paper | | Cas9/sgRNA expression vector with gRNA that targets 104 kb telomeric to *Sox2* TSS |
| Recombinant DNA reagent | pX330-Sox2-112T_gRNA | this paper | | Cas9/sgRNA expression vector with gRNA that targets 112 kb telomeric to *Sox2* TSS |
| Recombinant DNA reagent | pX330-Sox2-242T_gRNA | this paper | | Cas9/sgRNA expression vector with gRNA that targets 242 kb telomeric to *Sox2* TSS |
| Sequence-based reagent | Sox2 qPCR Forward Primer | this paper | | 5'-CTACGCGCACATGAACGG-3' |
| Sequence-based reagent | Sox2 qPCR Reverse Primer | this paper | | 5'-CGAGCTGGTCATGGAGTTGT-3' |
| Sequence-based reagent | Sox2 qPCR 129 allele Probe | this paper | | 5'-/56-FAM/CAACCGATG /ZEN/CACCGCTACGA/ 3IABkFQ/−3' |
| Sequence-based reagent | Sox2 qPCR Cast allele Probe | this paper | | 5'-/56-FAM/CAGCCGATG /ZEN/CACCGATACGA/ 3IABkFQ/−3' |
| Sequence-based reagent | Tbp qPCR Forward Primer | this paper | | 5'-ACACTCAGTTACAGGTGGCA-3' |
| Sequence-based reagent | Tbp qPCR Reverse Primer | this paper | | 5'-AGTAGTGCTGCAGGGTGATT-3' |
| Sequence-based reagent | Tbp qPCR Pan allele Probe | this paper | | 5'-/56-FAM/ACACTGTGT/ ZEN/GTCCTACTGCA/3IABkFQ/−3' |
| Sequence-based reagent | Genotyping PCR Primers | this paper | | see *Supplementary file 1* |
| Sequence-based reagent | CRISPR guide sequences | this paper | | see *Supplementary file 2* |
| Peptide, recombinant protein | Leukemia inhibitory factor (Lif) | Peprotech | 250–02 | |
| Peptide, recombinant protein | APC-Streptavidin | BD-Biosciences | 554067; RRID:AB_10050396 | Flow 1:200 |
| Peptide, recombinant protein | Insulin | Sigma | I6634 | |
| Peptide, recombinant protein | Epidermal growth factor (EGF) | Peprotech | 315–09 | |
| Peptide, recombinant protein | Fibroblast growth factor basic (Fgfb) | R and D Systems | 233-FB | |
| Peptide, recombinant protein | Natural mouse laminin | Thermo Fisher | 23017015 | |
| Peptide, recombinant protein | Bone morphogenetic protein 4 (BMP4) | R and D Systems | 314 BP | |
| Peptide, recombinant protein | Vascular endothelial growth factor (VEGF) | R and D Systems | 293-VE | |
| Peptide, recombinant protein | Activin A | R and D Systems | 338-AC | |
| peptide, recombinant protein | Fibroblast growth factor 10 (Fgf10) | R and D Systems | 345-FG | |
| Peptide, recombinant protein | Laminin-511 | iWaichem | N-892011 | |

*Continued on next page*

*Continued*

| Reagent type (species) or resource | Designation | Source or reference | Identifiers | Additional information |
|---|---|---|---|---|
| Chemical compound, drug | Prolong Live Antifade Reagent | Thermo Fisher | P36975 | |
| Chemical compound, drug | ascorbic acid | Sigma | A45-44 | |
| Chemical compound, drug | 1-thioglycerol | Sigma | M6145 | |
| Chemical compound, drug | PD03259010 | Selleckchem | S1036 | |
| Chemical compound, drug | CHIR99021 | Selleckchem | S2924 | |
| Chemical compound, drug | 5,6-Dichlorobenzimidazole 1-β-D-ribofuranoside | Sigma | D1916 | |
| Chemical compound, drug | JF646 | PMID: 28869757 | | |
| Software, algorithm | MS2Reporter AnalysisPipeline_knn Model.py | this paper | | Python scripts can be accessed on github (*Alexander, 2018*; copy archived at https://github.com/elifesciences-publications/2018_eLife_Alexander_et_al) |
| Other | Tetraspeck fluorescent beads | Thermo Fisher | T7279 | |
| Commerical assay, kit | KAPA Library Quantification Kit | Roche | KK4854 | |
| Commerical assay, kit | SPRIselect | Beckman Coulter | B23319 | |

## ESC Culture

129/CastEiJ F1 hybrid mouse embryonic stem cells were maintained in 2i + Lif media, composed of a 1:1 mixture of DMEM/F12 (Thermo Fisher Waltham, MA, #11320–033) and Neurobasal (Thermo Fisher #21103–049) supplemented with N2 supplement (Thermo Fisher #17502–048), B27 with retinoid acid (Thermo Fisher #17504–044), 0.05% BSA (Thermo Fisher #15260–037), 2 mM GlutaMax (Thermo Fisher #35050–061), 150 µM 1-thioglycerol (Sigma St. Louis, MO, M6145), 1 µM PD03259010 (Selleckchem Houston, TX, #1036), 3 µM CHIR99021 (Selleckchem #S2924) and $10^6$ U/L leukemia inhibitory factor (Peprotech Rocky Hill, NJ, #250–02). Media was changed daily and cells were passaged every 2 days.129/CastEiJ ESCs were genetically verified by PCR amplification and Sanger sequencing of regions within the *Sox2* locus to identify predicted SNPs between the parental genomes. These cells tested negatively for mycoplasma using MycoAlert Detect Kit (Lonza Basal, Switzerland #LT07-318).

## ESC genome modification

For insertion of PhiC31 and Bxb1 attP sequences, 150,000 cells were electroporated with 1 µM of single-stranded oligonucleotide donor containing the attP sequence and 400 ng of the sgRNA/Cas9 dual expression plasmid pX330 (a gift from Feng Zhang, Addgene Plasmid #42230) using the Neon Transfection System (Thermo Fisher). Neon settings for the electroporation were as follows: 1400V, 10 ms pulse width, three pulses. Electroporated ESCs were given 3 days to recover, followed by seeding approximately 5000 cells on a 10 cm dish for clone isolation (see Clone Isolation).

For integration of the tetO and cuO array, 150,000 cells were electroporated with 300 ng each of (1) a tetOx224 repeat plasmid bearing a PhiC31 attB sequence and a FRT-flanked neomycin resistance cassette, (2) a cuOx144 repeat plasmid bearing a Bxb1 attB sequence and a floxed puromycin or blasticidin resistance cassette, (3) an expression plasmid for the PhiC31 integrase (a gift from Philippe Soriano, Addgene Plasmid #13795), and (4) an expression plasmid for the Bxb1 integrase using the Neon Transfection System. Electroporated ESCs were allowed to recover for 3 days, followed by 7 days of drug selection using 500 μg/mL G418 and either 1 μg/mL puromycin or 8 μg/mL blasticidin in antibiotic-free media. After drug selection, cells were electroporated again with 400 ng each of Cre and Flpo expression plasmids to remove the resistance cassettes. 3 days after electroporation, approximately 5000 cells were seeded on a 10 cm plate for clone isolation (see Clone Isolation).

For targeting of the MS2 reporter construct into the endogenous *Sox2* allele, we generated a targeting plasmid that inserted a P2A sequence followed by the puromycin resistance gene upstream of the endogenous *Sox2* stop codon with 1 kb homolog arms on either side. We next mutated the PAM sequence for our sgRNA in the 3' homology arm by site-directed mutagenesis. 24 repeats of the MS2 hairpin sequence were inserted into an EcoRI restriction site located just 3' of the puromycin stop codon. 150,000 cells were electroporated with 400 ng of targeting plasmid and 400 ng of pX330 expressing the appropriate sgRNA. Electroporated ESCs were given 3 days to recover, followed by 5 days of puromycin selection. Approximately 5000 cells were subsequently seeded on a 10 cm dish for clone isolation (see Clone Isolation). A positive clone was identified by PCR. DNA sequencing confirmed no mutations in the *Sox2*-P2A-puro$_r$ cassette and identified a single bp deletion in the 3' UTR of the non-targeted CastEiJ allele due to residual targeting of a non-canonical NAG PAM.

For deletion of the *Sox2* Control Region or the Sox2-1-112T fragment, 150,000 cells were electroporated with 400 ng each of pX330 expressing sgRNAs targeting genomic regions centromeric and telomeric to the deletion fragment. 3 days after electroporation, approximately 5000 cells were seeded on a 10 cm plate for clone isolation (see Clone Isolation).

## ESC clone isolation

After 5–6 days of growth at low density (~5000 cells per 10 cm dish), individual colonies were picked and transferred to a 96-well plate. Briefly, colonies were aspirated and transferred to a well with trypsin, followed by quenching and dissociation with 2i + Lif + 5% FBS. Once the 96-well plate had grown to confluency, we split the clones into 2 identical 96-well plates. One plate was frozen at −80°C by resuspending the clones in 80% FBS/20% DMSO freezing media. The second plate was used for DNA extraction. All wells were washed once with PBS and subsequently lysed overnight at 55°C in a humidified chamber with 50 μL lysis buffer (10 mM Tris-HCl, pH 8.3, 50 mM KCl, 1.5 mM MgCl$_2$, 0.45% NP40, 0.45% Tween 20, 100 μg/mL Proteinase K). Genomic DNA was concentrated by ethanol precipitation and resuspended in 100 μL of double distilled water. 1 μL of suspension was used for subsequent PCR screening reactions using GoTaq Master Mix (Promega Madison, WI, #M7123).

## Stable expression of fluorescent transgenes

To generate stable lines expressing CymR, TetR, and MS2cp fluorescent protein fusions, 150,000 cells were electroporated with 400 ng of an ePiggyBac Transposase expression plasmid (a gift from Ali Brivanlou) and 50 ng of expression plasmid bearing PiggyBac terminal repeats. 7 days after electroporation, fluorescent cells were resuspended in fluorescence-activated cell sorting (FACS) buffer (5% FBS in PBS) and purified via FACS using a FACSAria II (BD). To enrich cells expressing the CymR-Halox2 fusion protein, ESCs were incubated in 100 nM of Janeila Fluor 646 (a gift from Luke Lavis) for 30 min at room temperature, washed once in FACS Buffer, incubated for 30 min at room temperature in FACS Buffer, washed again, and sorted using a FACSAria II.

## Isolation of Neural Progenitor Cells from ESCs

ESCs were passaged onto gelatinized 6 wells at 50,000–100,000 cells. The following day, these cultures were switched to N2B27 media (1:1 composition of DMEM/F12 and Neurobasal, N2 supplement, B27 with retinoic acid, 0.05% BSA, 2 mM GlutaMax, 150 μM 1-thioglycerol, 25 μg/mL insulin (Sigma #I6634)). After 4 days, we dissociated the cultures and seeded 1 million cells in an ungelatinized 10 cm dish in N2B27 with 10 ng/mL FGF basic (R and D Systems Minneapolis, MN, #233-FB)

and 10 ng/mL EGF (Peprotech #315–09) to form neurospheres. After 3–4 days of outgrowth, neuro-spheres were collected by gentle centrifugation (180xg, 3 min) and plated onto a pre-gelatinized six well. Neural progenitor cell (NPCs) lines were established by passaging (4–6 passages). For maintainance of NPCs, cells were cultured on wells pre-treated with poly-D-lysine and 4 µg/mL natural mouse laminin (Thermo Fisher #23017015) in N2B27 with 10 ng/mL FGF basic and 10 ng/mL EGF and passaged every 4–5 days.

## Differentiation of NPCs to neurons and astrocytes

To differentiate NPCs to astrocytes, 30,000 cells were plated onto coverglass within a 24 well pre-treated with poly-D-lysine and laminin. The following day, cells were switched to N2B27 with 10 ng/mL BMP4 (R and D Systems #314 BP) and allowed to differentiate for 12 days.

To differentiate NPCs to neurons, 30,000 cells were plated onto coverglass within a 24 well pre-treated with poly-D-lysine and laminin. The following day, cells were switched to N2B27 with 10 ng/mL FGF basic and allowed to differentiate for 6 days. Cells were then switched to N2B27 without additional factors and grown for 6 days.

## Differentiation of cardiogenic mesodermal precursors from ESCs

ESCs were dissociated and seeded to form embryoid bodies at 1 million cells per dish in SFD media (3:1 composition of IMDM (Thermo Fisher #12440–053) and Ham's F12 (Thermo Fisher #11765–054), N2 supplement, B27 without retinoic acid (Thermo Fisher #12587–010), 0.05% BSA, 2 mM GlutaMax, 50 µg/mL ascorbic acid (Sigma #A-4544), 450 µM 1-thioglycerol). After 2 days, EBs were dissociated and reaggregated at 1 million cells per dish in SFD media with 5 ng/mL VEGF (R and D Systems #293-VE), 5 ng/mL Activin A (R and D Systems #338-AC), and 0.75 ng/mL BMP4 to induce cardiogenic mesoderm. 40 hr after induction, cells were dissociated and stained for Flk1 and PDGFRα. Briefly, cells were washed four times in FACS Buffer, followed by incubation for 30 min with a biotinylated anti-FLK-1 antibody (Hybridoma Clone D218, 1:100). Cells were then washed three times with FACS Buffer and incubated with a PE-conjugated anti-PDGFRα (Thermo Fisher #12-1401-81, 1:400) and APC-Streptavidin (BD Biosciences Franklin Lakes, NJ, #554067, 1:200) for 30 min at room temperature. Cells were then washed two times with FACS Buffer and sorted for FLK1$^+$/PDGFRα$^+$ cells.

## Immunofluorescence

NPCs or differentiated astrocytes/neurons on coverglass were fixed for 10 min at room temperature with 4% paraformaldehyde in PBS. After fixing, the coverglass were washed twice with PBS, permeabilized in PBS with 0.5% Triton X-100 for 10 min, and washed once in PBS with 0.1% Triton. Cells were then blocked for 1 hr at room temperature in PBS/0.1% Triton/4% goat serum. After blocking, coverglass were incubated in primary antibody in PBS/0.1% Triton/4% goat serum overnight at 4°C in a humidified chamber. Coverglass were subsequently washed three times with PBS/0.1% Triton and incubated in secondary antibody in PBS/0.1% Triton/4% goat serum at room temperature for 1 hr. After secondary incubation, coverglass were washed three times with PBS/0.1% Triton, stained with DAPI in PBS (1 µg/mL), and mounted on a slide for imaging in mounting medium (1x PBS, pH7.4, 90% glycerol, 5 mg/mL propyl gallate). Antibodies used were anti-SOX2 (Santa Cruz Biotechnology Dallas, TX, #sc-365823, Lot# K1414), anti-PAX6 (Biolegend San Diego, CA, #901301, Lot# B235967), anti-TUBB3 (Biolegend #801201, Lot# B199846), and anti-GFAP (Sigma #G3893, Lot# 105M4784V).

## Western blotting

3 million cells were collected, washed once with PBS, and lysed in 4x Laemmli Buffer. Cell lysate was passed through a 30 gauge needle twenty times to shear the genomic DNA and the lysate was cleared by centrifugation at 13,000 RPM for 10 min at 4°C. Subsequently, lysate was supplemented with 100 mM DTT and boiled at 95°C for 10 min. 200,000 cells of protein lysate were loaded onto a Bis-Tris 4–12% polyacrylamide gel (ThermoFisher #NW04120BOX) and electrophoresis was carried out using the Bolt system (ThermoFisher). Protein was transferred to a PVDF membrane. Membranes were blocked for 1 hr at room temperature with 4% milk PBS Tween (PBST). Membrane was subsequently incubated in primary antibody overnight in 4% milk PBST at 4°C. Membranes were then

washed four times 15 min at room temperature in PBST and incubated in secondary antibody in 4% milk PBST for 1 hr at room temperature. After secondary incubation, membranes were washed four times 15 min at room temperature in PBST, incubated in SuperSignal chemiluminescence HRP substrate (ThermoFisher #34075), and visualized by film exposure. Antibodies used were anti-SOX2 (Santa Cruz #sc-365823, Lot# K1414) and anti-β-actin (Abcam Cambridge, UK, ab8227, Lot# GR92448-1).

## Quantitative PCR

RNA was extracted from 500,000 to 1,000,000 million cells using TRIzol and 200 ng of RNA was reversed transcribed using the QuantiTect Reverse Transcription kit (Qiagen Hilden, Germany). Quantitative PCR was performed on 8 ng cDNA in technical triplicates using TaqMan Gene Expression Master Mix (ThermoFisher #4369016) on a 790HT Fast Real-Time PCR System (ThermoFisher). The primer and probe sets used are as follows:

*Sox2* Forward primer – 5'CTACGCGCACATGAACGG3',
*Sox2* Reverse primer – 5'CGAGCTGGTCATGGAGTTGT3',
*Sox2* 129 allele probe –/56-FAM/CAACCGATG/ZEN/CACCGCTACGA/3IABkFQ/,
*Sox2* CastEiJ allele probe –/56-FAM/CAGCCGATG/ZEN/CACCGATACGA/3IABkFQ/, Tbp Forward primer – 5'ACACTCAGTTACAGGTGGCA3',
Tbp Reverse primer – 5'AGTAGTGCTGCAGGGTGATT3',
Tbp probe -/56-FAM/ACACTGTGT/ZEN/GTCCTACTGCA/3IABkFQ.
56-FAM = Fluorescein
ZEN = internal quencher (IDT)
3IABkFQ = 3' Iowa Black quencher

## Circular chromosome conformation capture (4C) Sequencing

4C using the *Sox2* promoter as a bait region was prepared as previously described (*van de Werken et al., 2012*). Primers used for 4C amplification are as follows:

*Sox2* promoter Forward primer - CAAGCAGAAGACGGCATACGAGATACXXXXXXGTGAC TGGAGTTCAGACGTGTGCTCTTCCGATCTGAATTAGGGGTTGAGGACAC
*Sox2* promoter Reverse primer – AATGATACGGCGACCACCGAGATCTACACTCTTTCCC TACACGACGCTCTTCCGATCTAGAGGGTAATTTTAGCCGATC

where XXXXXX stands for a barcode sequence and sequence complementary to the viewpoint fragment containing the *Sox2* promoter is underlined.

Single cell suspensions of mouse embryonic stem cells were cross-linked with 1% formaldehyde in PBS for 10 min at room temperature. Nuclei were isolated in lysis buffer (10 mM Tris-HCl pH8.0, 10 mM NaCl, 0.2% Igepal CA630, 1X protease inhibitor),and cross-linked chromatin was digested with DpnII (0.4 U/μL, 100U total) overnight at 37°C. This was followed by proximity ligation with 2000 units of T4 DNA Ligase (NEB, #M0202, 2 U/μL) for 4 hr at room temperature. After ligation, samples were treated with 1 mg/mL Proteinase K, 10% SDS for 30 min at 55°C, followed by reverse crosslinking through addition of 5M NaCl and heating to 65°C overnight. Circularized DNA was then linearized by subsequent digestion with 50 units of NlaIII (0.1 U/μL) overnight at 37°C. Typically, 200 ng of the resulting 4C template was used for the subsequent PCR reaction. The 4C template was PCR amplified for 30 cycles and 3–4 reactions were pooled together. Primers were designed such that the single-end read would sequence the primer binding site of the bait region and read into the target region of interest. The primers were designed to include Illumina adaptor sequences as well as barcodes derived from Illumina's TruSeq adaptors, which allowed for multiplexing of 4C-seq reactions. The PCR products were then purified using dual SPRI bead selection (Beckman Coulter, Indianapolis, IN Cat# B23319) to get template between 120–1000 bp according to the manufacturer's instructions. The concentrations of each 4C library were calculated using the KAPA qPCR system (Roche Basal, Switzerland Cat# KK4854) and comparison to a standard curve. The libraries were then combined and sequenced on a HiSeq 4000 (Illumina San Diego, CA) with single-end 50 bp reads.

For generating near-cis plots, 4C reads were first trimmed using cutadapt (*Martin, 2011*, RRID: SCR_011841) to remove the reading primer sequences, then mapped to the mm9 genome using bwa (*Li and Durbin, 2009*, RRID:SCR_010910). Mapped reads were filtered for valid 4C fragments,

normalized to reads per million, and visualized at the *Sox2* genomic locus using Basic4CSeq (*Walter et al., 2014*, RRID:SCR_002836).

For allele-specific read assignments, 4C reads were trimmed using cutadapt. Reads were then mapped to a modified mm9 genome using bowtie2 (*Langmead and Salzberg, 2012*) with default settings, where base positions annotated as heterozygous in the F123 129/CastEiJ hybrid cell line were masked. Reads mapping within the SCR (mm9 genomic coordinates chr3: 34653927–34660927) were then assigned to the 129 or Cast allele using SNPsplit.

## Live-Cell microscopy

We imaged all experiments on a Nikon Ti-E microscope and the following setup for live, spinning disk confocal microscopy: Yokogawa CSU-22 spinning disk, 150 mW Coherent OBIS 488 nm laser, 100 mW Coherent OBIS 561 nm laser, 100 mW Coherent OBIS 640 nm laser, a Yokogawa 405/491/561/640 dichroic, zET405/488/561/635 m quad pass emission filter, Piezo Z-drive, Okolab enclosure allowing for heating to 37°C, humidity control, and $CO_2$ control, and a Plan Apo VC 100x/1.4 oil immersion objective. Image acquisition utilized either a Photometric Evolve Delta EMCCD or an Andor iXon Ultra EMCCD camera. Pixel size using this set up was 91 nm.

ESCs were plated one day prior to imaging on a 8-chambered coverglass (VWR Radnor, PA, #155409) pretreated for at least 2 hr with 3.1 µg/mL Laminin-511 (iWaichem Tokyo, Japan #N-892011) at 120,000 cells per chamber. Just prior to imaging, 2i + Lif media was pre-mixed with 50 µg/mL ascorbic acid and a 1:100 dilution of Prolong Live Antifade Reagent (ThermoFisher P36975). If the cells to be imaged also expressed CymRHalox2, 100 nM of JF646 was also added to the media. After a one hour incubation, we added this media to the ESCs to be imaged. When indicated, ESC media was supplemented with 75 µM DRB (Sigma D1916) and incubated for 1 hr prior to image acquisition.

NPCs were plated at least 8 hr prior to imaging on a 8-chambered coverglass pre-treated with poly-D-lysine and laminin at 120,000 cells per chamber. Prior to imaging, N2B27 with FGF basic and EGF was pre-mixed with 50 µg/mL ascorbic acid and a 1:100 dilution of Prolong Live Antifade Reagent. After a one hour incubation, we added this media to the NPCs to be imaged.

Cardiogenic mesodermal cells enriched by FACS for FLK1 and PDGFRα were plated on 8-chambered coverglass precoated with 0.1% gelatin in StemPro-34 (Thermo Fisher #10639–011) supplemented with 2 mM GlutaMax, 50 µg/mL ascorbic acid, 5 ng/mL VEGF, 10 ng/mL FGF basic, and 25 ng/mL FGF10 (R and D Systems #345-FG) and cultured for 24 hr. Just prior to imaging, StemPro-34 media (with the additives listed above) was supplemented with a 1:100 dilution of Prolong Live Antifade Reagent, incubated for one hour, and subsequently added to the cultures for imaging.

For imaging experiments using CymRGFP and TetRtdTom, we captured green and red images by toggling the 488 nm and 561 nm lasers with a zET405/488/561/635 m multi-band pass emission filter in place, respectively. All colors were collected per plane prior to moving to next the plane (i.e. Z1-C1, Z1-C2, Z2-C1, Z2-C2, etc.) Z-planes were spaced 300 nm apart and exposure times were 30 ms. Each z-stack was composed of 21–28 slices and spaced 20 s apart. A single z-stack using this protocol required approximately 1.6 s for completion. Examples of raw and denoised data stacks used for analysis can be found at DOI: 10.5281/zenodo.2658814; https://zenodo.org/record/2658814#.XNDLAhNKjyw.

For imaging experiments using CymRHalox2-JF646, TetRGFPx2, abd tdMS2cp-tagRFP-Tx2, we imaged the green and far red channels as above except the the 488 nm and 640 nm lasers were used. After completion of the initial z-stack, a second z-stack was constructed at identical z-positions using the 561 nm laser and a ET525/50 m emission filter to capture tdMS2cp-tagRFP-Tx2 fluorescence. This eliminated bleed-through signal from the JF646 dye during 561 nm excitation allowed by the quad pass emission filter. Exposure times for this second z-stack were 50 ms.

All images were acquired using µManager (*Edelstein et al., 2010*, RRID:SCR_016865). Imaging data for each condition is composed of a minimum of three imaging sessions, except for cardiogenic mesodermal cultures, in which duplicate differentiations were performed.

## Image processing

Images were background subtracted using a dark image, converted to 32-bit, and denoised using NDSafir (*Carlton et al., 2010*; *Kervrann and Boulanger, 2006*) with the following settings:

ndsafir_priism input_image denoised_image -4d = zt -noise="poisson' -iter = 4 -p = 1 -sampling=-1 -adapt = 10 -island = 4 usetmp.

Denoised images were reverted back to 16-bit, fluorescence bleach corrected using exponential fitting and despeckled to remove high-frequency noise using ImageJ (*Schindelin et al., 2012*; *Schneider et al., 2012*, RRID:SCR_003070).

## Image analysis

### Tracking loci

Maximum Z-projections of 3D time series were manually analyzed to identify cuO/CymR and tetO/TetR spots in nuclei and annotate individual loci as doublets (likely two sister chromatids) or singlets. Loci that showed any frames with doublet spots for either channel were not included in downstream analysis. For each *Sox2* locus with well-behaved singlets, an ROI was drawn that included the locus location throughout the timecourse (or if the locus became untrackable due to leaving the field of view, the duration of its visibility). In some cases (e.g. NPCs), multiple ROIs were needed to track a single loci because of large-scale movements of the cell nucleus. In these cases, location data were merged together after tracking. For each locus, the 3D location for the cuO/CymR spot and the tetO/TetR spot was tracked within the delimited ROI using TrackMate (*Tinevez et al., 2017*) and its Laplacian of Gaussian spot detector with a sparse LAP tracker. The following additional settings were used:

    DO_SUBPIXEL_LOCALIZATION = true
    RADIUS = 2.5 pixels
    THRESHOLD = 0
    DO_MEDIAN_FILTERING = false
    ALLOW_TRACK_SPLITTING = false
    ALLOW_TRACK_MERGING = false
    LINKING_MAX_DISTANCE = 20 pixels for ESCs, MES/40 pixels for NPCs
    GAP_CLOSING_MAX_DISTANCE = 30 pixels for ESCs, MES/60 pixels for NPCs
    MAX_FRAME_GAP = 3 frames
    LINKING_FEATURE_PENALTIES = (QUALITY: 1.0, POSITION_Z: 0.8)
    GAP_CLOSING_FEATURE_PENALTIES = (QUALITY: 1.0, POSITION_Z: 0.8)
    TRACK_FILTER = TRACK_DURATION: 10

A Spot Quality Filter was also applied to result in detection of 20% more spots than the number of frames in the time course. This threshold was found to minimize spurious spot detection while also minimizing the loss of bona-fide cuO/tetO localization. In the case where solely the location of the Sox2 promoter was of interest (i.e. for identifying and quantitating *Sox2* transcriptional bursts across all cells), cuO/CymR-Halox2 spots were tracked as above except ALLOW_TRACK_MERGING was set to true. This facilitated recording a single track when the *Sox2* locus showed two sister chromatids.

TrackMate tracks for each spot were manually inspected, and if multiple tracks existed (due to gaps in the tracking), these were merged through manual curation. Spot positions converted to physical distances using a 0.091 µm pixel size and a 0.3 µm z-step.

### Correction for chromatic aberration

We corrected for chromatic aberration by collecting a single z-stack of TetraSpeck fluorescent beads (ThermoFisher #T7279) embedded in 2% agrose using the 488 nm, 561 nm, and 640 nm laser. Exposure time for was 50 ms. Positions of the beads were determined using TrackMate and its Laplacian of Gaussian spot detector with the following additional settings:

    DO_SUBPIXEL_LOCALIZATION = true
    RADIUS = 2.5 pixels
    THRESHOLD = 0
    DO_MEDIAN_FILTERING = false
    SPOT_FILTER = QUALITY: 100

Differences between the position of each bead in the green and red as well as green and far-red channel were determined. Based on these data, we calculated linear models for chromatic

aberration correction in X, Y, and Z based on position within the field of view. The following corrections were applied to the green channel when being compared to red:

$$X_{corrected} = 0.00027X_{raw} + 0.00728$$

$$Y_{corrected} = 0.00028Y_{raw} - 0.00303$$

$$Z_{corrected} = -0.00139Z_{raw} - 0.1954$$

The following corrections were applied to the green channel when being compared to far-red:

$$X_{corrected} = -0.0005X_{raw} + 0.02553$$

$$Y_{corrected} = -0.00044Y_{raw} + 0.01949$$

$$Z_{corrected} = -0.00325Z_{raw} - 0.15869$$

## Localization precision estimation

Tetraspeck (Thermo Fisher T7279) multicolor fluorescent beads were embedded in 2% agarose and a one hundred frame Z-stack time series was constructed at various laser intensities. The max spot intensity as well as the mean and standard deviation of the nuclear background was estimated from ten nuclei for both cuO/CymR and tetO/TetR using our raw time-lapse data. Bead time series were modified to add background noise using ImageJ to approximate the nuclear background and then denoised as described above. 9–15 beads that showed signal within one standard deviation of that observed for either the cuO/CymR or tetO/TetR spots were tracked using TrackMate and the following additional settings:

    DO_SUBPIXEL_LOCALIZATION = true
    RADIUS = 2.5 pixels
    THRESHOLD = 0
    DO_MEDIAN_FILTERING = false

A spot filter for spot quality was set manually to only include the top 100 detected spots. The standard deviation of position of each bead was computed in the X, Y, and Z dimensions using a five frame sliding window to generate a distribution of estimated uncertainties.

As an addition measure of localization precision, *Sox2*-SCR ESCs expressing CymR-GFP and TetR-tdTom were cultured on coverglass as described for live-cell imaging above. Prior to imaging, the growth medium was removed and cells were fixed using 4% PFA in PBS for 5 min at room temperature. Cells were washed once with PBS and imaged in PBS. Two color z-stacks were captured at 72 time points using settings that were identical to live-cell microscopy except that there was no time delay between frames. These images were then processed identically to that used for live-cell microscopy. Spots from 10 to 14 *Sox2* loci were tracked across the time course using TrackMate and the following settings:

    DO_SUBPIXEL_LOCALIZATION = true
    RADIUS = 2.5 pixels
    THRESHOLD = 0
    DO_MEDIAN_FILTERING = false

A spot filter for spot quality was set manually to only include the top 72 detected spots. As with the fluorescent beads, the standard deviation of position for both the cuO and tetO label was computed in the X, Y, and Z dimensions using a five frame sliding window to generate a distribution of estimated uncertainties.

## Simulation of 3D distance measurement bias and uncertainty

To estimate the measurement bias of distance measurements, 1000 X, Y, and Z positions were sampled from normal distributions with standard deviations equal to the median values of our localization precision estimates (X = 12 nm, Y = 10 nm, Z = 36 nm for cuO and X = 16 nm, Y = 16 nm,

Z = 50 nm for tetO). The means of these distributions were fixed a 0 for cuO and varied over a range for tetO to simulate a range of separation distances. True distance was calculated as the Euclidean distance between points located at the center of the cuO and tetO distributions. Simulated measured distance was taken as the mean of the sampled Euclidean distances.

To estimate the uncertainty of distance measurements, we repeated the analysis above except the number of sampled positions was increased to 50,000. Simulated distance uncertainty was taken as the interquartile range of the simulated measured distances.

## Euclidean distance
1D, 2D, and 3D euclidean distances were calculated using the formula:

$$Dist_{ij} = \sqrt{\sum_{v=1}^{n} (X_{vi} - X_{vj})^2}$$

where $i$ and $j$ represent the cuO/CymR and tetO/TetR spot, respectively, and $n$ the number of dimensions.

## Relative displacement
The relative position of spot1 (CymRGFP) with respect to spot2 (TetRtTom) for the $v$th dimension was calculated as follows:

$$X_{v\hat{i}} = (X_{vi} - X_{vj})$$

The relative displacement was then calculated as the change is the relative position of spot 1.

$$Disp_t = \sqrt{\sum_{v=1}^{n} (X_{v\hat{i}}(t) - X_{v\hat{i}}(t-1))^2}$$

where $t$ is the current frame and $n$ is the number of dimensions.

## Autocorrelaton analysis
Autocorrelation values were calculated according to the formula

$$A(\tau) = \frac{E[(D_t - \mu)(D_{t+\tau} - \mu)]}{\sigma^2}$$

where $D_t$ represents distance at time $t$, $\tau$ is the time lag, $\mu$ and $\sigma^2$ are the average and variance of 3D distance measured across the cell population, and $E$ is the expected (i.e. average) value. Confidence intervals were computed by bootstrapping and recalculating $A(\tau)$ across 1000 simulations to estimate 95% confidence.

## Distribution distances and clustering
The distance between 3D distance probability distributions from two cell lines or cell types was computed using earth mover's distance (EMD). Briefly, the earth mover's distance is the minimum cost to convert one probability distribution to the other over a defined region. We calculated pairwise EMD for each 3D probability distribution using the R package *earthmovdist*. Complete-linkage hierachical clustering was then performed to generate a dendrogram.

## MS2 signal identification and quantification
3D time-lapse images of tdMS2cp-tagRFP-Tx2 were converted into 2D images by maximum Z projection. For each *Sox2* locus considered for analysis, a 20 × 20 pixel region centered on the XY tracking position of the cuO/CymR spot, reflecting the position of the *Sox2* promoter region, was analyzed for each frame. If tracking information was missing for a given frame, the position coordinates from the nearest frame were used. This 20 × 20 region was used for parameter estimation for 2D Gaussian fitting using the equation:

$$f(x,y) = Ae^{-\left(\frac{(x-x_o)^2}{2\sigma_x^2} + \frac{(y-y_o)^2}{2\sigma_y^2}\right)} + C$$

where $A$ (Gaussian height), $x_o, y_o$ (location of Gaussian peak), $\sigma_x^2, \sigma_y^2$ (Gaussian variance), and $C$(offset) are all estimated parameters. Initial estimate of the offset was defined as the median pixel value in the ROI, $A$ was estimated as the maximum pixel value minus the estimated offset, $\sigma_x^2$ and $\sigma_y^2$ were estimated as 1, and $x_o, y_o$ was estimated as the location of the brightest pixel in the ROI. These initial estimates were used attempt a Gaussian fit on a 10x10 pixel region centered on the estimated Gaussian position. We constrained the potential Gaussian fit to have a minimum height of 10% above background fluorescence, a fit position of no more than 3 pixels from the estimated position, and a width of no more than 4. Successful Gaussian fits were filtered for likelihood to reflect true MS2 signal using a k-nearest neighbor model trained on manually classified data and 4 parameters of the fit ($A$, $\sigma_x^2$, $\sigma_y^2$, and an $R^2$ value). Furthermore, frames were also required to have at least one neighboring frame ($\pm$ 3 frames) also demonstrate MS2 signal, eliminating high frequency noise. Time points which passed these filter steps were assigned a relative MS2 Signal based on:

$$Signal = \frac{A + C}{Normalization\ Factor}$$

were the normalization factor was the median pixel value for the 20 $\times$ 20 pixel ROI across all time-points. For time points that did not pass filter, MS2 signal was taken as the median value of the 20 $\times$ 20 ROI for the current frame normalized as above.

## Sox2 burst classification

*Sox2* burst initiation events were classified as frames positive for MS2 signal (see above) that lack MS2 positive classifications for the preceding three frames. All frames spanning the burst initiation and the last positive MS2 classification prior to the next burst initiation were labeled as one burst event.

## Aligned Sox2 burst analysis

To align our MS2 data across all *Sox2* bursts, a defined window was sampled for each burst centered on the burst initiation event. We subsequently generated a randomly sampled control comparison for this analysis by randomly shuffling the frames labeled as burst initiation events and repeating the analysis. Mean distances or MS2 signal were then calculated based on relative frame compared to the burst initiation event. Confidence intervals were computed by bootstrapping and recalculating the mean value for each relative frame across 1000 simulations to estimate 95% confidence.

## Browser tracks

Unless wiggle files were available as part of the accession, sequencing read archives (SRA) were downloaded from NCBI and reads were aligned to the mm9 mouse genome using Bowtie (*Langmead et al., 2009*, RRID:SCR_005476) as part of the Galaxy platform (*Afgan et al., 2018*, RRID:SCR_006281). Sequences were extended by 200 bp and allocated into 25 bp bins to generate wiggle files. HiC data were visualized using JuiceBox (*Durand et al., 2016*). Browser tracks were visualized on the UCSC Genome browser (*Kent et al., 2002*, RRID:SCR_005780).

## Acknowledgements

We thank Elphege Nora, Geoffrey Fudenberg, Brian Black, and members of the Lomvardas and Weiner labs for helpful discussion; Elphege Nora, Benoit Bruneau, and Kirstin Meyer for a critical reading of the manuscript; Lena Bengtsson for experimental help and technical assistance; and Edith Heard, Patrick Devine, Feng Zhang, Elphege Nora, Michele Calos, Robert Singer, Philippe Soriano, Barbara Panning, Ali Brivanlou and Luke Lavis for helpful reagents. This work was supported by an American Heart Association Postdoctoral Fellowship (#16POST309100006, JMA); NIH Grants R21EB022787 (ODW), R35GM118167 (ODW), R01DC013560 (SL), T32GM007175 (MS), R21HG010065 (YS),

UM1HG009402 (YS), and R21EB021453 (BH); and the WM Keck Foundation Medical Research Grant (BH). BH is a Chan Zuckerberg Biohub Investigator.

## Additional information

### Funding

| Funder | Grant reference number | Author |
| --- | --- | --- |
| American Heart Association | 16POST309100006 | Jeffrey M Alexander |
| National Institute of General Medical Sciences | R35GM118167 | Orion D. Weiner |
| National Institute on Deafness and Other Communication Disorders | R01DC013560 | Stavros Lomvardas |
| National Institute of Biomedical Imaging and Bioengineering | R21EB022787 | Orion D. Weiner |
| National Institute of Biomedical Imaging and Bioengineering | R21EB021453 | Bo Huang |
| W. M. Keck Foundation | Medical Research Grant | Bo Huang |
| National Human Genome Research Institute | R21HG010065 | Yin Shen |
| National Human Genome Research Institute | UM1HG009402 | Yin Shen |
| National Institute of General Medical Sciences | T32GM007175 | Michael Song |

The funders had no role in study design, data collection and interpretation, or the decision to submit the work for publication.

### Author contributions

Jeffrey M Alexander, Conceptualization, Data curation, Formal analysis, Investigation, Visualization, Methodology, Writing—original draft, Writing—review and editing; Juan Guan, Software, Methodology, Writing—review and editing; Bingkun Li, Formal analysis, Visualization, Writing—review and editing; Lenka Maliskova, Investigation, Writing—review and editing; Michael Song, Formal analysis, Investigation, Visualization, Writing—review and editing; Yin Shen, Orion D Weiner, Conceptualization, Supervision, Funding acquisition, Writing—review and editing; Bo Huang, Supervision, Writing—review and editing; Stavros Lomvardas, Supervision, Funding acquisition, Writing—review and editing

### Author ORCIDs

Jeffrey M Alexander (iD) https://orcid.org/0000-0002-2258-5738
Bo Huang (iD) https://orcid.org/0000-0003-1704-4141
Stavros Lomvardas (iD) http://orcid.org/0000-0002-7668-3026
Orion D Weiner (iD) https://orcid.org/0000-0002-1778-6543

### Decision letter and Author response

Decision letter https://doi.org/10.7554/eLife.41769.053
Author response https://doi.org/10.7554/eLife.41769.054

# Additional files

## Supplementary files

• Supplementary file 1. Protocol for insert of cuO-/tetO-arrays into mouse ESCs. Protocols for targeting the cuO and/or tetO array(s) into genomic regions of interest in mouse ESCs.
DOI: https://doi.org/10.7554/eLife.41769.024

• Supplementary file 2. Primer sequences used in cell line characterization. List of PCR primer sequences and expected amplicon size used in the study. Brief description of the purpose of each primer pair is included.
DOI: https://doi.org/10.7554/eLife.41769.025

• Supplementary file 3. 20 bp guide RNA sequences used in CRISPR/Cas9 genome engineering. List of 20 bp sequences homologous to the mouse 129 genome designed into CRISPR/Cas9 sgRNAs. Targeted genomic location (mm9 coordinates), genome strand, and brief description of purpose for sgRNA is included.
DOI: https://doi.org/10.7554/eLife.41769.026

• Supplementary file 4. Data table from 3D tracking of cuO/CymR and tetO/TetR labels. All data used in the study for cuO/CymR and tetO/TetR localization. C1 refers to Channel 1 (cuO/CymR). C2 refers to Channel2 (tetO/TetR). For examples of raw and denoised data files that were used for this analysis, see DOI: 10.5281/zenodo.2658814; https://zenodo.org/record/2658814#.XNDLAhNKjyw. Columns are as follows:

**Cell_Line**– label used to identify cell line
**Batch**– unique microscopy session identifier
**C1_T_Step-sec**– step size between frames
**Locus_ID**– unique identifier for each Sox2 locus
**C1_TrackID**– track identifier from TrackMate
**C1_Track_Length**– track length from TrackMate
**C1_SpotID**– spot identifier from TrackMate
**C1_X_Value_pixel** – X position in pixels for C1 spot
**C1_Y_Value_pixel** – Y position in pixels for C1 spot
**C1_Z_Value_slice** – Z position in slices for C1 spot
**C1_T_Value_frame** – frame of measurement
**C1_X_Value_um** – X position in microns for C1 spot
**C1_Y_Value_um** – Y position in microns for C1 spot
**C1_Z_Value_um** – Z position in microns for C1 spot
**C1_T_Value_sec** – time point in seconds for measurement
**C2_TrackID**– track identifier from TrackMate
**C2_Track_Length**– track length from TrackMate
**C2_SpotID**– spot identifier from TrackMate
**C2_X_Value_pixel** – X position in pixels for C2 spot
**C2_Y_Value_pixel** – Y position in pixels for C2 spot
**C2_Z_Value_slice** – Z position in slices for C2 spot
**C2_T_Value_frame** – imaging frame
**C2_X_Value_um** – X position in microns for C2 spot
**C2_Y_Value_um** – Y position in microns for C2 spot
**C2_Z_Value_um** – Z position in microns for C2 spot
**C2_T_Value_sec** – time point in seconds
**X_Distance_um**– X distance between C1 and C2 labels
**Y_Distance_um**– Y distance between C1 and C2 labels
**Z_Distance_um**– Z distance between C1 and C2 labels
**XY_Distance_um**– XY distance between C1 and C2 labels
**XYZ_Distance_um**–XYZ distance between C1 and C2 labels,
**C1_Corrected_X_Value**_um – X position in microns for C1 spot after correcting for chromatic aberration,
**C1_Corrected Y_Value_um**–Y positfion in microns for C1 spot after correcting for chromatic aberration

**C1_Corrected Z_Value_um**–Z position in microns for C1 spot after correcting for chromatic aberration
**Corrected_X_Distance_um**–X distance after correcting for chromatic aberration
**Corrected_Y_Distance_um** – Y distance after correcting for chromatic aberration
**Corrected_Z_Distance_um** – Z distance after correcting for chromatic aberration
**Corrected_XY_Distance_um** – XY distance after correcting for chromatic aberration
**Corrected_XYZ_Distance_um** – XYZ distance after correcting for chromatic aberration
**Relative_C1_Corrected_X_Value_um**–X position of C1 label relative to the position of C2 in microns
**Relative_C1_Corrected_Y_Value_um**–Y position of C1 label relative to the position of C2 in microns
**Relative_C1_Corrected_Z_Value_um**–Z position of C1 label relative to the position of C2 in microns
**Relative_XY_Displacement_um**–Relative XY distance traveled by the C1 label between adjacent frames
**Relative_XYZ_Displacement_um**–Relative XYZ distance traveled by the C1 label between adjacent frames
**Relative_XY_Angle_radians**–Relative angle between two successive displacements for the C1 label in the XY plane

DOI: https://doi.org/10.7554/eLife.41769.027

• Supplementary file 5. Data table for MS2 transcription analysis for all loci. All data used in transcriptional analysis of *Sox2* locus. Columns are as follows:
**Cell_Line**– label used to identify cell line
**Locus_ID**– unique identifier for each Sox2 locus
**Gauss_Filter**– whether the MS2 Gaussian fit passed the knn filter step
**Noise_Filter**–whether the MS2 Gaussian fit passed a high frequency noise filter step
**Pass_Filter**–whether the MS2 signal for the given frame was classified as transcriptional signal. Required both Gauss_Filter = TRUE and Noise_Filter = TRUE
**Gaussian_Height_Threshold**–minimum relative height above background allowed for Gaussian fit
**Gaussian_Width_Threshold**–maximum Gaussian variance allowed for Gaussian fit
**Background**–Offset calculated from Gaussian fit. If no Gaussian fit was found, set to median pixel intensity of ROI
**Gaussian Height**–Amplitude calculated from Gaussian fit. If no Gaussian fit was found, set to 0
**Gaussian_Volume**–Volume under fitted Gaussian. If no Gaussian fit was found, set to 0
**Local_Median**–Median pixel intensity of ROI
**Norm_MS2_Signal**–Relative height of MS2 gaussian normalized to background. For frames that did not pass filter, local median value was used in pace of gaussian height. See MATERIALS and METHODS for more details.
**R_Squared**–Coefficient of determination between 2D gaussian fit and experimental data
**T_Value_frame**– imaging frame
**X_Value_pixel**– X position in pixels for C2 spot (cuO/CymR)
**X_Location**– X position of peak of fit Gaussian
**X_Sigma**– X dimension variance of fit Gaussian
**Y_Value_pixel**– Y position in pixels for C2 spot (cuO/CymR)
**Y_Location**– Y position of peak of fit Gaussian
**Y_Sigma**– Y dimension variance of fit Gaussian
**Z_Value_slice**– Z position in slices for C2 spot (cuO/CymR)
**Batch**– unique microscopy session identifier.

DOI: https://doi.org/10.7554/eLife.41769.028

• Supplementary file 6. Data table for MS2 transcription analysis and 3D localization for Sox2-SCR Singlets. Data used to compare transcriptional activity of *Sox2* locus to 3D distances between *Sox2* and SCR. C1 refers to Channel 1 (tetO/TetR). C2 refers to Channel2 (cuO/CymR). Columns are as in *Supplementary files 3* and *4* with one additional column: Active_Transcribing– Whether the locus demonstrated any MS2 signal that passed filter during imaging session.

DOI: https://doi.org/10.7554/eLife.41769.029

• Supplementary file 7. Data table of atatistical comparison of distances centered on transcriptional bursts. Summary statistics and associated Mann-Whitney p-values for pairwise comparisons between burst centered and random centered distances.
DOI: https://doi.org/10.7554/eLife.41769.030

• Transparent reporting form
DOI: https://doi.org/10.7554/eLife.41769.031

### Data availability

All microscopy localization data utilized in this study are included as supplementary files. Example raw confocal stacks and denoised confocal stacks from Batch65 imaging available on Zenodo. Tracking data for cuO and tetO from these images can be found in Supplementary file 4. Details of microscopy acquisition in Materials and Methods. Sequencing data have been deposited in GEO under accession code GSE127901 and SRA under accession code PRJNA523665.Python scripts can be accessed on GitHub at https://github.com/JMAlexander/2018_eLife_Alexander_et_al (copy archived at https://github.com/elifesciences-publications/2018_eLife_Alexander_et_al).

The following datasets were generated:

| Author(s) | Year | Dataset title | Dataset URL | Database and Identifier |
|---|---|---|---|---|
| Alexander JM, Guan J, Li B, Maliskova L, Song M, Shen Y, Huang B, Lomvardas S, Weiner OD | 2019 | 4C on Sox2 Locus with tetO/cuO Modifications | https://www.ncbi.nlm.nih.gov/geo/query/acc.cgi?acc=GSE127901 | NCBI Gene Expression Omnibus, GSE127901 |
| Alexander JM, Guan J, Li B, Maliskova L, Song M, Shen Y, Huang B, Lomvardas S, Weiner OD | 2019 | Live-Cell Imaging Reveals Enhancer-dependent Sox2 Transcription in the Absence of Enhancer Proximity | https://doi.org/10.5281/zenodo.2658814 | Zenodo, 10.5281/zenodo.2658814 |

The following previously published datasets were used:

| Author(s) | Year | Dataset title | Dataset URL | Database and Identifier |
|---|---|---|---|---|
| Wamstad JA, Alexander JM, Truty RM, Shrikumar A, Li F, Ellertson KE, Ding H, Wylie JN, Pico AR, Capra JA, Erwin G, Kattman SJ, Keller GM, Srivastava D, Levine SS, Pollard KS, Holloway AK, Boyer LA, Bruneau BG | 2013 | ChIP-seq analysis of histone modifications and RNA polymerase II at 4 stages of directed cardiac differentiation of mouse embryonic stem cells | https://www.ncbi.nlm.nih.gov/geo/query/acc.cgi?acc=GSE47949 | NCBI Gene Expression Omnibus, GSE47949 |
| Vierstra J, Rynes E, Sandstrom R, Thurman RE, Zhang M, Canfield T, Sabo PJ, Byron R, Hansen RS, Johnson AK, Vong S, Lee K, Bates D, Neri F, Diegel M, Giste E, Haugen E, Dunn D, Humbert R, Wilken MS, Josefowicz S, Samstein R, Chang K, Levassuer D, Disteche C, De Bruijn M, Rey TA, Skoultchi A, Ru- | 2014 | Mouse regulatory DNA landscapes reveal global principles of cis-regulatory evolution | https://www.ncbi.nlm.nih.gov/geo/query/acc.cgi?acc=GSE51336 | NCBI Gene Expression Omnibus, GSE51336 |

| | | | | | |
|---|---|---|---|---|---|
| densky A, Orkin SH, Papayannopoulou T, Treuting P, Selleri L, Kaul R, Bender MA, Groudine M, Stamatoyannopoulos JA | | | | | |
| Chen X, Xu H, Yuan P, Fang F, Huss M, Vega VB, Wong E, Orlov YL, Zhang W, Jiang J, Loh YH, Yeo HC, Yeo ZX, Narang V, Govindarajan KR, Leong B, Shahab A, Ruan Y, Bourque G, Sung WK, Clarke ND, Wei CL, Ng HH | 2008 | Mapping of transcription factor binding sites in mouse embryonic stem cells | https://www.ncbi.nlm. nih.gov/geo/query/acc. cgi?acc=GSE11431 | NCBI Gene Expression Omnibus, GSE11431 | |
| de Wit E, Vos ES, Holwerda SJ, Valdes-Quezada C, Verstegen MJ, Teunissen H, Splinter E, Wijchers PJ, Krijger PH, de Laat W | 2015 | CTCF binding polarity determines chromatin looping | https://www.ncbi.nlm. nih.gov/geo/query/acc. cgi?acc=GSE72539 | NCBI Gene Expression Omnibus, GSE72539 | |
| Bonev B, Mendelson Cohen N, Szabo Q, Fritsch L, Papadopoulos G, Lubling Y, Xu X, Lv X, Hugnot J, Tanay A, Cavalli G | 2017 | Multi-scale 3D genome rewiring during mouse neural development | https://www.ncbi.nlm. nih.gov/geo/query/acc. cgi?acc=GSE96107 | NCBI Gene Expression Omnibus, GSE96107 | |
| Creyghton MP, Cheng AW, Welstead GG, Kooistra T, Carey BW, Steine EJ, Hanna J, Lodato MA, Frampton GM, Sharp PA, Boyer LA, Young RA, Jaenisch R | 2010 | ChIP-Seq of chromatin marks at distal enhancers in Mouse Embryonic Stem Cells and adult tissues. | https://www.ncbi.nlm. nih.gov/geo/query/acc. cgi?acc=GSE24164 | NCBI Gene Expression Omnibus, GSE24164 | |
| Zhang Y, Wong CH, Birnbaum RY, Li G, Favaro R, Ngan CY, Lim J, Tai E, Poh HM, Wong E, Mulawadi FH, Sung WK, Nicolis S, Ahituv N, Ruan Y, Wei CL | 2013 | Chromatin connectivity maps reveal dynamic promoter-enhancer long-range associations | https://www.ncbi.nlm. nih.gov/geo/query/acc. cgi?acc=GSE44067 | NCBI Gene Expression Omnibus, GSE44067 | |
| Hansen AS, Pustova I, Cattolico C, Tjian R, Darzacq X | 2017 | CTCF and cohesion regulate chromatin loop stability with distinct dynamics | https://www.ncbi.nlm. nih.gov/geo/query/acc. cgi?acc=GSE90994 | NCBI Gene Expression Omnibus, GSE90994 | |

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
