## [Decision Letter]

Thank you for submitting your article "Live-cell imaging reveals enhancer-dependent *Sox2* transcription in the absence of enhancer proximity" for consideration by *eLife*. Your article has been reviewed by three peer reviewers, and the evaluation has been overseen by a Reviewing Editor and Kevin Struhl as the Senior Editor. The following individual involved in review of your submission has agreed to reveal his identity: Zhe Liu (Reviewer #3).

The reviewers have discussed the reviews with one another and the Reviewing Editor has drafted this decision to help you prepare a revised submission.

The work provides the first precise measurement of enhancer-promoter distances correlated with transcriptional output. Surprisingly, the proximity of the enhancer and promoter was not correlated with the output. This challenges current dogma that these two elements must come together to initiate transcription. The authors propose an activator hub model where factors are concentrated around the promoter in a large volume.

While the reviewers are positive about the work (one even thinks it is a "landmark"), they have a number of concerns, mainly centering in the accuracy of the distance measurements. Since these distances are critical to the argument and since the conclusion is heterodox, confidence in the measurements are essential. For instance, how was chromatic aberration corrected? They suggest additional controls to verify the measurements. An example would be to show the tagging did not affect the looped interactions by comparing 3C interactions between the experimental and wild type cells. Additional suggestions were to improve the text for clarity and accuracy in interpretation.

Please read through the suggestions for improvements and determine whether you can respond adequately within a few months. If so we will entertain a revised manuscript for review.

Reviewer #1:

This manuscript investigates the long held belief that distal enhancer sites must directly contact the promoter site to activate transcription. This required heroic genetic labeling of the endogenous *Sox2* promoter, a *Sox2* enhancer locus, as well as the messenger RNA being transcribed. They conclude by many analyses that the position separation between enhancer and promoter is not correlated with gene expression in contrast to the expectation of a direct contact between promoter and enhancers. They propose that recent observation of condensates of transcription factors, and or more complex delay models may be at play in the *Sox2* locus investigated. This is an experimental tour de force that will prove important, prompt and guide many future experiments. The manuscript warrants publication in *eLife*. There are questions on the accuracy of the position measurements that should be addressed as a major concern as that will set a standard for how future measurements may be done; however these should be doable within the time frame allowed by the journal for a revision.

A major issue to be resolved is the question of how accurately the chromatin labeling arrays represent the *Sox2* promoter and SCR positions. This becomes apparent when comparing the MS2 signal position to the *Sox2* promoter marker position in 3 color imaging. The MS2 signal typically appears to be detected far from the *Sox2* promoter signal. This may be due to technical reasons (different filter cube, time delay between *Sox2*/SCR and MS2 stacks) or represent actual spatial separation between the promoter chromatin label and the *Sox2* gene position. This is particularly true since a 14kb separation in the deletion mutant appears to result in a typical separation of ~250nm and never below 100nm. The authors should provide a control for the positional accuracy of their chromatin labels with respect to the target sequence, e.g. co-staining of the actual target locus by DNA FISH or dCas9-based chromatin labeling.

Reviewer #2:

In this paper, Alexander and co-workers address the important topic and enhancer-promoter (E-P) contacts using the *Sox2* gene in mESCs as a model. While there was a recent E-P live-cell imaging study in *Drosophila* from the Gregor group, the Gregor system was a bit artificial and genome organization is very different between mammals and flies. The present study by Alexander is therefore very important: To my knowledge, it is the first live-cell imaging study of E-P contacts in mammals. This is important, because Hi-C, which averages over cell populations and only generates a snapshot cannot readibly report on dynamics. Getting at the dynamics can only be achieved with live-cell imaging, which is what Alexander has now accomplished.

Other highlights include a nice general system for tagging DNA loci (though authors need to put plasmids on AddGene), nice controls (e.g. the other cell lines with similar distances and the 111 kb deletion), comparing mESCs and NPCs and the simultaneous MS2-readout to simultaneously look at transcription.

The findings are also surprising and will be of wide interest. I believe there is a strong possibility that this paper will be looked back upon in a couple of years as a landmark paper in the field and I believe it will be of very wide interest.

Nevertheless, I have a series of serious technical concerns, which should be addressed and I believe that authors should do one important control experiment: verify using a "C"-method that the E-P loop is not disrupted. Finally, given the technical concerns – some of which may not be fully addressable – the authors need to more clearly state the limitations of their work in the main text. Also, many imaging details that are crucial, are missing from the Materials and methods.

Activator hub model

In Figure 6H, authors propose an "activator hub model" where a large hub (maybe 200-400 nm?) activates over long distance. This is an interesting model. If it is true that it is so big, presumably many other genes would be inside of it. Are there other other genes within 1 Mb of *Sox2* on the same chromosome? Are they ON or OFF and if some of them are OFF, how do they stay OFF if there is a large hub?

If the hub is a 400 nm cube and mouse ES cells are diploid, they should have 2 of these hubs and around 50k genes (since diploid). Using the typical volume of a nucleus (e.g. 8 μm cube), one gets total hub volume 0.128 μm3 and nuclear volume of 512 μm3, corresponding to 12.5 genes inside of the hubs. Is this realistic? Numbers chosen here are a bit random, but the point is that it seems a bit dangerous to have a large hyper-activating hub in the nucleus (like the LLPS studies the authors reference) since it would be likely to randomly contact genes that should be OFF – especially since chromatin moves around as the authors show. If this hub lasts for 10 minutes, how many random genes will bump into it? The nucleus is a pretty crowded environment. Can the authors discuss this a bit more clearly?

New tools to visualize DNA loci should be on AddGene.

In addition to the biological insight, a big impact of this paper will be the new tools the authors develop to insert TetO and CuO sites in the genome. The 2-step modular approach with attP, PhiC31, Bxb1 etc. is clever and the TetO and CuO plasmids will be generally useful. However, I could not find the AddGene Accession codes for these vectors. In the revised manuscript, the authors should deposit these plasmids to AddGene and include the accession numbers in the manuscript. Moreover, the authors should write a brief protocol on how to use the plasmids and attach it to the manuscript. This will greatly increase the impact of the paper and serve as a big positive contribution to the community.

Key control

It is very nice that authors verify that array insertion does not affect *Sox2* expression according to qPCR. This is a really important control. However, the missing and equally important control is the verification that the *Sox2*-SCR looping interaction is not affected. Authors could argue that since SCR is required for expression, the fact that *Sox2* qPCR is the same, suggests that looping level is not affected. But since the authors suggest that E-P loops don't directly affect transcription, this is no longer the case. Therefore, an essential (and straightforward) control experiment to do for the revised manuscript is a 3C-qPCR (or another C-type) experiment comparing *Sox2*-SCR E-P contacts in WT cells, cells with the arrays but without TetR and CymR and cells with arrays and also TetR and CymR.

Localization Precision

I am somewhat skeptical of the localization precision. It seems a bit weird that the X and Y values are so different. Also 10-15 nm is really high precision. It seems almost too good. I worry that even if the authors tried to use beads at lower light intensity, this could bias the calculation. It is also not clear how well a TetraSpeck bead approximates the unknown distribution of in vivo conformations of e.g. an 8 kb array inside a live cell. Is there any way the authors can use the TetO and CuO readouts to estimate the errors? E.g. in fixed cells?

Distance between Promoter and SCR and CuO and TetO arrays

The distance between the *Sox2* E and P is quite high (17 kb). I totally get that it is tricky: if you put the arrays too close, they may interfere with function. If you put them too far away, they may not be good reporters and it is not obvious to me what the best distance would be. But given the wide distribution in Figure 2C yellow line, I believe the authors should emphasize a bit more in the main text that this introduces some uncertainty and is an important caveat.

Timescale of E-P loop and time-scale for MS2 appearance

One key thing I was missing was a discussion of the time-scale of E-P loops. E.g. recently there have been papers arguing that CTCF/cohesin loops are either stable or dynamic and it would be nice if the authors could discuss how their observations relate to this (even if they do not directly observe discrete E-P loops). For example, does the *Sox2* loop occur inside a CTCF/Cohesin loop and can the authors compare to some of the CTCF/Cohesin timescales?

Along these lines, the analysis in Figure 6 is very important in that it tries to find a correlation between E-P distance and transcription. But although the result is negative, can the authors really exclude that E-P contact is necessary for *Sox2* transcription.

Suppose the following scenario: E-P loops form and last for 10 seconds (but duration highly stochastic, sometimes 1 sec sometimes 100 sec). Soon after they break, *Sox2* E and P move apart and the distance increases. The E-P loop even when the true distance is <50 nm, will show a broad distribution of distances similar to yellow line in Figure 2C. After E-P contact, Transcription factors, histone modifying enzymes, mediator, Brd4, p300, TBP, SAGA, TFIID and other factors are recruited but sequentially and with delay between each. This takes an unknown amount of time. Then Pol 2 is recruited. Pol 2 pauses for a bit and then begins transcribing. Since the MS2 reporter is 3', there is a very long delay between Pol 2 initiation and MS2/MCP-readout (the authors should calculate the expected time it takes from initiation to MS2 appearance using the estimated Pol II elongation speed and the length of the *Sox2* modified gene and report this duration in the main text). For the sake of argument, let's say this process takes 7 min on average, but because of the many steps, each of which is stochastic, the duration is broadly distributed and heavy tailed such that it can take anywhere from 3 min to 15 min (or something like this).

In this scenario with: 1) very transient E-P contact measured using the very high localization uncertainty shown by the yellow line in Figure 2E; 2) highly stochastic and variable duration for in-between steps and 3) long and somewhat variable delay before MS2 appearance since reporter is 3' and 4) E-P contacts only produce transcription burst say 40% of the time. Would the authors really be able to detect a positive correlation using the analysis in Figure 6?

My sense is that the authors could not, though I would be happy to be persuaded otherwise by a careful quantitative analysis. This does not mean that the author's contribution is not highly valuable, but unless they can exclude this possibility, they should state explicitly in the main text or discussion that they cannot exclude that their analysis fails to detect the underlying E-P inducing *Sox2* transcription.

Authors kind of sketch this in 6H top panel, but I found the discussion about these limitations unclear and lacking. It is much better to clearly state the limitations.

Encounter definition

Authors include a very nice control cell line, where 111 kb has been deleted between the pairs. This cell line is "always in encounter" in the sense that the CuO and TetO arrays are about as close as they would be in a bona-fide E-P loop. Looking at Figure 2C, it looks like the mean distance is 250 nm and the range is approximately 0-500 nm. That means that perfect E-P co-localization can nevertheless appear as 500 nm at low probability. But in Figure 4E, authors define encounter as 100 nm. If the mean E-P distance during an encounter is 250 nm (Figure 2C), defining the threshold to be 100 nm seems too restrictive.

Obviously, it is very interesting and informative to consider the probability of an encounter during a time window, but given the much larger mean distance for the control E-P loop cell line, 100 nm is too small. I am not sure how best to deal with this but current Figure 4E seems unfair.

One option would be for the authors to clearly state this limitation in the main text and then re-plot Figure 4E for multiple thresholds – e.g. 100, 150, 200, 250, 300, 350 nm. At the very least, they should also consider thresholds a little bigger than the mean E-P distance in the control cell line (yellow line in Figure 2C).

Information about imaging and the microscope

Technical information about the microscope and imaging protocol is extremely important to evaluate the study, but highly lacking.

What was the pixel size?

What were the emission filters?

How many z-stacks and how long exposure times?

What were the time-gaps between z-stacks? What was the physical distance between the z-stack?

I could not understand – did the authors collect all colors per plane and then move to the next plane or did authors do sequential all planes for each color and then acquire next color?

Authors must report duration of z-stacks?

How did authors correct for chromatic aberrations? Authors mention shifting position, but I could not understand what they did.

How did authors align color channels?

How did authors determine 3D positions? Was it PSF-fitting? If so, what was the PSF-model? Did they enforce symmetric XY PSF or allow asymmetric XY-PSF? Did they do MLE or LS fitting?

What were the settings used in TrackMate? Were gaps allowed?

Etc. Etc. Please provide all details in the Materials and methods since they are important.

Reviewer #3:

Large Picture:

A central model in the current understanding of gene regulation is that direct physical interactions between promoter and enhancer are required for transcriptional activation. It is widely believed that long-distance promoter-enhancer communications are realized in the form of chromatin looping. In this manuscript, the authors devise comprehensive live-cell imaging experiments to test this model using *Sox2* locus and *Sox2* control region (SCR) – a strong long-distance enhancer required for *Sox2* expression. Specifically, by incorporating CuO and TetO arrays into *Sox2* locus and SCR respectively, authors established a robust molecular imaging system to precisely quantify the physical distance between *Sox2* and SCR in the nucleus of single living ES cells. Surprisingly, authors find no evidence supporting *Sox2*-SCR interactions in comparison with SCR-control loci pairs. And, consistent with DNA-FISH results on HoxD locus (Genes and Dev. 2014. 28: 2778-2791), authors also showed that upon ES cell differentiation, the genomic region containing *Sox2* and SCR compacts as the distance between *Sox2* and SCR becomes shorter. Most strikingly, author found no temporal correlation between *Sox2* transcription bursting and the proximity of the SCR to *Sox2* locus. These emerging results began to challenge the central model regarding DNA looping as the primary mechanism that mediates long-distance enhancer-promoter communications.

I found that the experiments done by authors are very well controlled and the results are timely for the field to move forward in search for alternative mechanisms. I would like to support the publication of the manuscript.

[Editors' note: further revisions were requested prior to acceptance, as described below.]

Thank you for resubmitting your work entitled "Live-cell imaging reveals enhancer-dependent *Sox2* transcription in the absence of enhancer proximity" for further consideration at *eLife*. Your revised article has been favorably evaluated by Kevin Struhl (Senior Editor), a Reviewing Editor, and three reviewers.

The manuscript is acceptable but there are some remaining issues that need to be addressed. In short, the reviewers would like to see some additional comments regarding the lateral resolution and the inclusion of some of the raw data.

Reviewer #1:

I think given the time frame the authors appropriately addressed the concerns raised.

The authors should provide some of the raw uncorrected video.

Reviewer #2:

We are satisfied with the revisions. This was a valiant effort, and the authors satisfied most of the requests. We believe that manuscript is vastly enriched.

One remaining issue it the claimed lateral resolution. Here the authors use a published method to estimate it that is providing a result that seems out of scale compared to what is typically achieved. I could not pinpoint what is wrong and therefore cannot comment more. Perhaps one way for the authors to explain their number would be to comment and provide their own rational comparing this value to other papers and explain how they improved it so much. Also it would be important to comment on the validity of the tool they use to localize a locus. This is important since other papers might use similar methods and will not necessarily achieve such results.

Because the lateral resolution issue will be a major problem to other labs trying to reproduce the data I would suggest that the authors prepare a set of raw videos that are representative of the dataset and their quantification of it so that others can use it as a benchmarking tool.

Reviewer #3:

This was a strong paper and, in the revision, authors have improved the manuscript more and satisfactorily addressed my concerns. I would support the publication of the paper in *eLife*.

---

## [Author Response]

Please read through the suggestions for improvements and determine whether you can respond adequately within a few months. If so we will entertain a revised manuscript for review.Reviewer #1:This manuscript investigates the long held belief that distal enhancer sites must directly contact the promoter site to activate transcription. This required heroic genetic labeling of the endogenous Sox2 promoter, a Sox2 enhancer locus, as well as the messenger RNA being transcribed. They conclude by many analyses that the position separation between enhancer and promoter is not correlated with gene expression in contrast to the expectation of a direct contact between promoter and enhancers. They propose that recent observation of condensates of transcription factors, and or more complex delay models may be at play in the Sox2 locus investigated. This is an experimental tour de force that will prove important, prompt and guide many future experiments. The manuscript warrants publication in eLife. There are questions on the accuracy of the position measurements that should be addressed as a major concern as that will set a standard for how future measurements may be done; however these should be doable within the time frame allowed by the journal for a revision.A major issue to be resolved is the question of how accurately the chromatin labeling arrays represent the Sox2 promoter and SCR positions. This becomes apparent when comparing the MS2 signal position to the Sox2 promoter marker position in 3 color imaging. The MS2 signal typically appears to be detected far from the Sox2 promoter signal. This may be due to technical reasons (different filter cube, time delay between Sox2/SCR and MS2 stacks) or represent actual spatial separation between the promoter chromatin label and the Sox2 gene position. This is particularly true since a 14kb separation in the deletion mutant appears to result in a typical separation of ~250nm and never below 100nm. The authors should provide a control for the positional accuracy of their chromatin labels with respect to the target sequence, e.g. co-staining of the actual target locus by DNA FISH or dCas9-based chromatin labeling.

We previously performed DNA FISH for the *Sox2* locus using standard protocols and bacterial artificial chromosome (BAC) probes to validate the position of our repressor arrays. However, the reviewer’s suggested experiment is significantly more challenging because the regions being probed are only kilobases in size. We have attempted the DNA FISH experiments proposed by the reviewer in collaboration with the Huang lab, who have significant DNA FISH expertise, but we have thus far been unsuccessful in generating sufficient signal-to-noise for the small probes that this experiment demands. We expect the time necessary to optimize and troubleshoot the proposed experiments would push significantly beyond the window suggested for timely resubmission. Similarly, the reviewer’s suggestion of using dCas9 to label the *Sox2* promoter or SCR is not feasible given the time constraints. Because the regions of interest are non-repetitive, 26-36 individual sgRNAs would need to be stably expressed in our ESC lines. These assays require extensive optimization and would not be possible to complete by the suggested deadline for resubmission. However, a number of the reviewers concerns can be addressed/explained without additional experiments.

The referenced discrepancy between the *Sox2* promoter position and MS2 transcriptional bursts is likely to be explained by how the imaging was performed. Due to the presence of 3 distinct fluorophores in these cells and spectral overlap between tagRFP-T and JF646, fluorescence data for the GFP and JF646 channels were captured by first sweeping through z-positions with a multi-bandpass zET405/488/561/635m emission filter and toggling the 488nm and 640nm lasers at each z-plane to build up a 2-color z-stack for the position labels (i.e. cuO and tetO). Once this initial z-stack was completed, a distinct emission filter (ET610/60m) was placed in the light path, and an additional z-stack at identical z-positions was collected using the 561nm laser to capture tagRFP-T fluorescence. This optical configuration prevented bleedthrough from the JF646 fluorescence into the tagRFP-T images. However, this imaging setup resulted in substantial delays (~ 5 second) between measuring tagRFP-T fluorescence and GFP/JF646 for a given z-plane. Furthermore, the change in emission filter modifies the light path for tagRFP-T images as compared to GFP/JF646. Together, these factors call for caution in directly comparing the precise positional information gleaned from an MS2cp-tagRFP-T spot and, for instance, cuO/CymRHalo, though they are sufficiently accurate to constrain the region of the nucleus in which we probe for transcriptional bursts. These technical considerations are likely major contributors to the differences pointed out of the reviewer.

The central point of the reviewer’s concern is that there is some uncertainty regarding how well our labels report on the precise positions of the relevant genomic regions (e.g. tetO on the SCR and cuO on the *Sox2* promoter). The most critical issue is how well the distances we measure relate to actual genomic separations of the probed regions. To address this consideration, we have developed and reported data for an appropriate control, the *Sox2*-del-SCR cell line. The 3D distances measured from this cell line support that our experimental setup can detect differences between the *Sox2* locus and a “constitutively engaged” control. It also provides a measure of how closely the distance of our genetic labels (~14 kb combined from the enhancer and promoter) report on a configuration when the *Sox2*/SCR regions are in constitutive proximity (0 kb separation in the *Sox2*-del-SCR cell line and distance theoretically limited only by our localization precision). Thus, the separation distance for the *Sox2*-del-SCR ESC line gets at the underlying question raised by the reviewer. While the average separation value of ~250 nm in this line may appear large, we note that a study using CRISPR-Sirius to label repeats on human chromosome 19 report that the separation distance between two loci 4.6 kb apart demonstrated a wide range as well (20-350 nm, Ma et al., 2017, bioRxiv). It is true that due to the genetic distance between the cuO/tetO labels and the regions of interest (*Sox2*/SCR), our positional uncertainty in *Sox2*/SCR position is greater than that suggested by our localization precision (also discussed in response to reviewer 2, major comment 5). We have added text in the manuscript to better highlight these limitations (subsection “Visualization of the Sox2 Region in ESCs Reveals Minimal Evidence for Sox2/SCR Interactions”. However, these considerations do not compromise the major conclusions of the paper: 1) that the *Sox2*-SCR label pair shows no bias to proximity compared to equally-spaced controls and 2) that transcription show no bias towards *Sox2*-SCR proximity compared to non-transcribing time points.

Reviewer #2:In this paper, Alexander and co-workers address the important topic and enhancer-promoter (E-P) contacts using the Sox2 gene in mESCs as a model. While there was a recent E-P live-cell imaging study in Drosophila from the Gregor group, the Gregor system was a bit artificial and genome organization is very different between mammals and flies. The present study by Alexander is therefore very important: To my knowledge, it is the first live-cell imaging study of E-P contacts in mammals. This is important, because Hi-C, which averages over cell populations and only generates a snapshot cannot readibly report on dynamics. Getting at the dynamics can only be achieved with live-cell imaging, which is what Alexander has now accomplished.Other highlights include a nice general system for tagging DNA loci (though authors need to put plasmids on AddGene), nice controls (e.g. the other cell lines with similar distances and the 111 kb deletion), comparing mESCs and NPCs and the simultaneous MS2-readout to simultaneously look at transcription.The findings are also surprising and will be of wide interest. I believe there is a strong possibility that this paper will be looked back upon in a couple of years as a landmark paper in the field and I believe it will be of very wide interest.Nevertheless, I have a series of serious technical concerns, which should be addressed and I believe that authors should do one important control experiment: verify using a "C"-method that the E-P loop is not disrupted. Finally, given the technical concerns – some of which may not be fully addressable – the authors need to more clearly state the limitations of their work in the main text. Also, many imaging details that are crucial, are missing from the Materials and methods.Activator hub modelIn Figure 6H, authors propose an "activator hub model" where a large hub (maybe 200-400 nm?) activates over long distance. This is an interesting model. If it is true that it is so big, presumably many other genes would be inside of it. Are there other other genes within 1 Mb of Sox2 on the same chromosome? Are they ON or OFF and if some of them are OFF, how do they stay OFF if there is a large hub?If the hub is a 400 nm cube and mouse ES cells are diploid, they should have 2 of these hubs and around 50k genes (since diploid). Using the typical volume of a nucleus (e.g. 8 μm cube), one gets total hub volume 0.128 μm3 and nuclear volume of 512 μm3, corresponding to 12.5 genes inside of the hubs. Is this realistic? Numbers chosen here are a bit random, but the point is that it seems a bit dangerous to have a large hyper-activating hub in the nucleus (like the LLPS studies the authors reference) since it would be likely to randomly contact genes that should be OFF – especially since chromatin moves around as the authors show. If this hub lasts for 10 minutes, how many random genes will bump into it? The nucleus is a pretty crowded environment. Can the authors discuss this a bit more clearly?

We agree with the reviewer that the “activator hub model” leaves many unanswered questions. Perhaps most important, as the reviewer mentions, it is unclear how specificity in regulation would be achieved by such a mechanism. In addition, how long-lived these hubs persist and what factors would be involved are also important unknowns. We do not have good answers to these questions. It could be that additional binding events occur with specificity at the *Sox2* promoter and are essential to achieve regulated activation (i.e. the hub is permissive for activation). The reviewer’s concern that many genes could be spatially nearby and in danger of being non-specifically activated is reasonable. However, this scenario may be reduced by *Sox2*’s isolated genomic context. In mice, *Sox2*’s nearest protein-coding gene neighbors are ~570kb and 1.1Mb away in the centromeric and telomeric directions, respectively. Perhaps a hub mechanism is restricted to genes that are isolated within the genome. We don’t want to speculate too broadly in the text about all the factors of this model given that its is merely a hypothesis (for which we have no affirmatory data ourselves) and there are other potential models that are consistent with our observations. We have added some text to the Discussion that attempts to frame some of the important open questions in the hub model that the reviewer has raised (fourth paragraph).

Of interest, we have begun to investigate this mechanism by visualizing the mediator component Med1 using Med1-GFP reagents from Sabari et al., 2018. We do see several large (> 500 nm) and persistent (tens of minutes) aggregates of Med1-GFP fluorescence in the nucleus. Our very preliminary data suggests that these large aggregates are not enriched at the *Sox2* locus, regardless of transcription status. Thus, it does not appear that *Sox2* is activated by a very large hub of Mediator based on these findings. However, this is analysis for a single factor, and recent studies have suggested activators upstream (i.e. Brd4) and downstream (i.e. RNAP) of Mediator have this capacity as well. It will be interesting to similarly probe these other factors to determine if SCR is likely to communicate with *Sox2* via local concentration of activator proteins. However, to date, the “hub” model for SCR function is unvalidated.

New tools to visualize DNA loci should be on AddGene.In addition to the biological insight, a big impact of this paper will be the new tools the authors develop to insert TetO and CuO sites in the genome. The 2-step modular approach with attP, PhiC31, Bxb1 etc. is clever and the TetO and CuO plasmids will be generally useful. However, I could not find the AddGene Accession codes for these vectors. In the revised manuscript, the authors should deposit these plasmids to AddGene and include the accession numbers in the manuscript. Moreover, the authors should write a brief protocol on how to use the plasmids and attach it to the manuscript. This will greatly increase the impact of the paper and serve as a big positive contribution to the community.

We have deposited key plasmids for cuO and tetO targeting and CymR/TetR fusions on addGene. A detailed protocol for using the attP system to target the cuO and tetO arrays to the mouse genome in embryonic stem cells can be access on addGene along with these plasmids. This protocol has also been added as Supplementary File 1.

Key controlIt is very nice that authors verify that array insertion does not affect Sox2 expression according to qPCR. This is a really important control. However, the missing and equally important control is the verification that the Sox2-SCR looping interaction is not affected. Authors could argue that since SCR is required for expression, the fact that Sox2 qPCR is the same, suggests that looping level is not affected. But since the authors suggest that E-P loops don't directly affect transcription, this is no longer the case. Therefore, an essential (and straightforward) control experiment to do for the revised manuscript is a 3C-qPCR (or another C-type) experiment comparing Sox2-SCR E-P contacts in WT cells, cells with the arrays but without TetR and CymR and cells with arrays and also TetR and CymR.

We agree with the reviewer that this is an important control to include in our study. Therefore, we collaborated with the laboratory of Yin Shen, who have experience studying 3D contacts of the *Sox2* locus in embryonic stem cells using 3C-derivatives. With their help, we have included 4C analysis of *Sox2* promoter contacts in unmodified 129/Cast ESCs, ESCs with the cuO and tetO labels integrated adjacent to the *Sox2* promoter and SCR, respectively, and the cuO- and tetO-labeled cell line above that also expresses CymR-GFP and TetR-tdTom. We find that these modified cell lines still show clear evidence of the enriched *Sox2*-SCR contacts that have been previously described and that are present in the unmodified control. Furthermore, by using allele-specific polymorphisms within the SCR region, we find 4C signal at the SCR does not have allelic bias in our modified cell lines. The modified 129 allele contributes approximately half of the detected contacts between *Sox2* and SCR in our cell lines. These data demonstrate that *Sox2*-SCR contacts are not impacted in the cell lines used in to our study. These data have been incorporated into a new supplemental figure, Figure 1—figure supplement 3.

Localization PrecisionI am somewhat skeptical of the localization precision. It seems a bit weird that the X and Y values are so different. Also 10-15 nm is really high precision. It seems almost too good. I worry that even if the authors tried to use beads at lower light intensity, this could bias the calculation. It is also not clear how well a TetraSpeck bead approximates the unknown distribution of in vivo conformations of e.g. an 8 kb array inside a live cell. Is there any way the authors can use the TetO and CuO readouts to estimate the errors? E.g. in fixed cells?

We have performed localization analysis of fixed cells as the reviewer suggests to determine localization precision in a cellular context at the fluorescent levels achievable for our experiments. Briefly, *Sox2*-SCR ESCs were fixed in 4% paraformaldehyde for 5 minutes and subsequently imaged using the same microscopy setup and imaging conditions as live-cell microscopy. Images were denoised and processed as described for our live-cell datasets. Subsequently, 10-14 loci were tracked for 72 frames, and position uncertainty was estimated by calculating the standard deviation of positions in the X, Y, and Z dimensions for blocks of 5 consecutive measurements. This windowed approach helps minimize position uncertainty caused by stage drift throughout the imaging period. These analyses demonstrate the values reported using the fluorescent bead method may underestimate accuracy, as the median uncertainty (i.e. standard deviation) improved for all dimensions when using fixed cells. The fixed cell analysis may improve our localization precision due to residual local diffusion experienced by beads confined in a 2% agarose gel. See Author response image 1.

Thus, these new analyses corroborate the reported precision values of position localization and suggest the bead values may be conservative.

Distance between Promoter and SCR and CuO and TetO arraysThe distance between the Sox2 E and P is quite high (17 kb). I totally get that it is tricky: if you put the arrays too close, they may interfere with function. If you put them too far away, they may not be good reporters and it is not obvious to me what the best distance would be. But given the wide distribution in Figure 2C yellow line, I believe the authors should emphasize a bit more in the main text that this introduces some uncertainty and is an important caveat.

We agree that this is a confounding factor that we don’t emphasize enough in the original manuscript. We have added language in the main text that highlights this source of uncertainty (subsection “Visualization of the Sox2 Region in ESCs Reveals Minimal Evidence for Sox2/SCR Interactions”).

Timescale of E-P loop and time-scale for MS2 appearanceOne key thing I was missing was a discussion of the time-scale of E-P loops. E.g. recently there have been papers arguing that CTCF/cohesin loops are either stable or dynamic and it would be nice if the authors could discuss how their observations relate to this (even if they do not directly observe discrete E-P loops). For example, does the Sox2 loop occur inside a CTCF/Cohesin loop and can the authors compare to some of the CTCF/Cohesin timescales?

We feel a discussion of E-P looping time scales would be problematic in the current paper because our data does not provide robust insights regarding individual enhancer-promoter interactions. Thus, any discussion of these features would be based solely on the literature the reviewer mentions or require substantial speculation on our part. For instance, how long-lived might *Sox2*-SCR contacts be based on our data? This is difficult to say given that identifying time periods of interaction/contacts in our data has not been possible. Given these challenges (and the surprising nature of our observations), we have focused our discussion to features of the locus that we can directly measure (e.g. proximity).

Along these lines, the analysis in Figure 6 is very important in that it tries to find a correlation between E-P distance and transcription. But although the result is negative, can the authors really exclude that E-P contact is necessary for Sox2 transcription.Suppose the following scenario: E-P loops form and last for 10 seconds (but duration highly stochastic, sometimes 1 sec sometimes 100 sec). Soon after they break, Sox2 E and P move apart and the distance increases. The E-P loop even when the true distance is <50 nm, will show a broad distribution of distances similar to yellow line in Figure 2C. After E-P contact, Transcription factors, histone modifying enzymes, mediator, Brd4, p300, TBP, SAGA, TFIID and other factors are recruited but sequentially and with delay between each. This takes an unknown amount of time. Then Pol 2 is recruited. Pol 2 pauses for a bit and then begins transcribing. Since the MS2 reporter is 3', there is a very long delay between Pol 2 initiation and MS2/MCP-readout (the authors should calculate the expected time it takes from initiation to MS2 appearance using the estimated Pol II elongation speed and the length of the Sox2 modified gene and report this duration in the main text). For the sake of argument, let's say this process takes 7 min on average, but because of the many steps, each of which is stochastic, the duration is broadly distributed and heavy tailed such that it can take anywhere from 3 min to 15 min (or something like this).In this scenario with: 1) very transient E-P contact measured using the very high localization uncertainty shown by the yellow line in Figure 2E; 2) highly stochastic and variable duration for in-between steps and 3) long and somewhat variable delay before MS2 appearance since reporter is 3' and 4) E-P contacts only produce transcription burst say 40% of the time. Would the authors really be able to detect a positive correlation using the analysis in Figure 6?My sense is that the authors could not, though I would be happy to be persuaded otherwise by a careful quantitative analysis. This does not mean that the author's contribution is not highly valuable, but unless they can exclude this possibility, they should state explicitly in the main text or discussion that they cannot exclude that their analysis fails to detect the underlying E-P inducing Sox2 transcription.Authors kind of sketch this in 6H top panel, but I found the discussion about these limitations unclear and lacking. It is much better to clearly state the limitations.

It is difficult for our data for formally exclude such scenarios, because of the uncertainty regarding critical parameters in the reviewer’s hypothetical. For instance, depending on how many steps exist between E-P engagement and how long-lived (and variable) these steps are, such a model may be reconcilable with our observations. However, we believe that by orienting the community towards these types of models, our observations will greatly inform the conversation regarding enhancer mechanism of action. Our data provides the strongest evidence against a tight temporal coupling between E-P engagement and transcriptional activity. Indeed, this simple model is supported by the only other live-cell imaging study of enhancer-promoter communication (Chen et al., 2018), which demonstrates robust and immediate changes in distance upon transcription. Thus, it is important that our data suggest such a simple relationship between E-P contacts and transcription is unlikely to explain *Sox2* regulation. We are careful not to exclude the possibility that E-P contacts are involved in some capacity. Instead we emphasize that enhancer proximity (a parameter we can directly measure) do not correlate with transcription in time. Our data argue against a short, defined temporal lag between EP interactions and transcription (Figure 6E). We cannot rule out a very long or variable time lag between EP contacts and transcription from our analysis. We have added text within the Discussion (second paragraph) to expand on the reviewer’s comment and clarify how enhancer-promoter contacts could be involved in directing *Sox2* transcription.

Encounter definitionAuthors include a very nice control cell line, where 111 kb has been deleted between the pairs. This cell line is "always in encounter" in the sense that the CuO and TetO arrays are about as close as they would be in a bona-fide E-P loop. Looking at Figure 2C, it looks like the mean distance is 250 nm and the range is approximately 0-500 nm. That means that perfect E-P co-localization can nevertheless appear as 500 nm at low probability. But in Figure 4E, authors define encounter as 100 nm. If the mean E-P distance during an encounter is 250 nm (Figure 2C), defining the threshold to be 100 nm seems too restrictive.Obviously, it is very interesting and informative to consider the probability of an encounter during a time window, but given the much larger mean distance for the control E-P loop cell line, 100 nm is too small. I am not sure how best to deal with this but current Figure 4E seems unfair.One option would be for the authors to clearly state this limitation in the main text and then re-plot Figure 4E for multiple thresholds – e.g. 100, 150, 200, 250, 300, 350 nm. At the very least, they should also consider thresholds a little bigger than the mean E-P distance in the control cell line (yellow line in Figure 2C).

The purpose of this analysis is to explore how the slow rate of chromosomal conformation turnover observed at the *Sox2* locus would affect the availability of two chromosomal loci for specific interactions or encounters. The reviewer is correct that this is perhaps most interesting when thinking about an enhancer-promoter pair, but we are not explicitly testing the interaction frequency of the *Sox2* promoter and SCR element in this analysis.

This is because, as stated above, it has been difficult to identify individual E-P contacts from our data. Given this limitation, we have instead focused on how the observed dynamics of chromatin influence the encounter frequency of two chromosomal loci generally. In this context, we are interested in the encounter frequency of the tetO and cuO labels themselves rather than using these labels to report on *Sox2*/SCR encounter frequency. When considering the cuO and tetO label positions themselves, our uncertainty in their positions approaches our localization precision reported in Figure 2—figure supplement 2. The reviewer is correct that threshold for what we label an encounter is arbitrary, and the dependence that encounter frequency has on initial conformation should be robust to the selection of this threshold. We have added clarifying text for this section (subsection “Slow Sox2 Locus Conformation Dynamics Lead to Limited Exploration and Variable Enhancer Encounters”) in an attempt to better frame this analysis and have added the reviewer’s suggested analysis of multiple thresholds for each cuO/tetO label pair (Figure 4E). We have also included this analysis for the Control-Control and SCR-Control cell lines. We observed a consistent trend across label pairs and threshold values. See Author response image 2.

**Author response image 2. respfig2:** 

Information about imaging and the microscopeTechnical information about the microscope and imaging protocol is extremely important to evaluate the study, but highly lacking.

We have added additional details in the Materials and methods to include the important imaging details requested (sections Live-Cell Microscopy, Image Processing, and Image Analysis).

What was the pixel size?

91nm

What were the emission filters?How many z-stacks and how long exposure times?What were the time-gaps between z-stacks? What was the physical distance between the z-stack?

cuO/tetO: 30ms exposures, 300nm between slices, 21-28 slices in z-stack, 20s between time points

cuO/tetO/MS2: 30ms exposure (cuO/tetO) and 50ms exposure (MS2), 300nm between slices, 28-30 slices in z-stack, 30s between time points

I could not understand – did the authors collect all colors per plane and then move to the next plane or did authors do sequential all planes for each color and then acquire next color?

cuO/tetO: All colors were collected per plane prior to moving to next plane. That is Z1-C1, Z1-C2, Z2-C1, Z2-C2, etc..

cuO/tetO/MS2: Green and Far-red colors collected per plane prior to moving to next plane. Red (MS2) was then collected for each plane in second z-stack. That is Z1-C1, Z1-C3, Z2-C1, Z2-C3, …, Zlast-C1, Zlast-C3, Z1-C2, Z2-C2, …, Zlast-C2

Authors must report duration of z-stacks?

cuO/tetO: A single z-stack with 2 color imaging at 30ms exposures is completed in 1.63s.

How did authors correct for chromatic aberrations? Authors mention shifting position, but I could not understand what they did.

Chromatic aberration was a common concern from all reviewers. Thus, we have provided more information regarding how we adjusted our analysis to account for chromatic aberration below. We have also provided additional details in the Materials and methods section of the resubmitted manuscript.

We corrected for chromatic aberration by collecting a single z-stack of TetraSpeck fluorescent beads (ThermoFisher #T7279) embedded in 2% agrose using the 488 nm, 561 nm, and 640 nm laser. Positions of the beads were determined using TrackMate using the Laplacian of Gaussian spot detector. We then visualized these differences. In the plots in Author response image 3, the y-axis shows the difference between positions of the same bead in the red vs. green channel. The x-axis shows the position of the bead within the field of view (X and Y) or within the Z-stack (Z).

**Author response image 3. respfig3:** 

In most cases, we see the differences between red and green positions do not change much with position (the slope of the fit line is ~0), in which case the y-intercept gives the average offset due to chromatic aberration. From this, we can quickly see that chromatic aberration is most severe in the Z dimension. However, in some cases, we do observe position dependent effects. These were mostly found for dimensional shifts dependent on the bead location in that dimension (chromatic shift in X direction dependent on X position, etc.). Thus, we utilized the following linear models to apply a chromatic correction across our data.

Correction of X position

corrected_green_position (um) = (0.00027 * green_X_position (um) + 0.00728)

Correction of Y position

corrected_green_position (um) = (0.00028 * green_Y_position (um) – 0.00303)

Correction of Z position

corrected_green_position (um) = (-0.00139 * green_Z_position (um) – 0.1954)

Our data also allows us to perform a sanity check for the effectiveness of our chromatic aberration corrections. cuO-tetO distances in 1D space (X distance, Y distance, and Z distance) in an ideal system (no chromatic aberration) are expected to be normally distributed with a mean of 0. Thus, we can look to see if our corrections bring our measured cuO-tetO distance closer to this ideal. This is assessed in Author response image 4.

**Author response image 4. respfig4:** 

These corrections do well in recentering the data towards 0, validating our corrections.

Green and Far-Red Position Difference

We performed a similar analysis to determine the chromatic aberration between the green and far-red channels utilized for CymR-Halox2 and TetR-GFPx2 simultaneous imaging, Author response image 5.

**Author response image 5. respfig5:** 

As before, we fit linear models for each dimension to correct the green position for chromatic aberration relative to far-red.

Correction of X position

corrected_green_position (um) = (-0.0005 * green_X_position (um) + 0.02553)

Correction of Y position

corrected_green_position (um) = (-0.00044 * green_Y_position (um) + 0.01949)

Correction of Z position

corrected_green_position (um) = (-0.00325 * green_Z_position (um) – 0.15869)

We also evaluate how our corrected values change the calculated values for X, Y, and Z distances in Author response image 6.

**Author response image 6. respfig6:** 

Again, our corrections shift the distributions towards 0. We note these corrections are not as successful in eliminating deviations from 0 as the corrections applied to the CymRGFP/ TetR-tdTom datasets. This may derive from a larger stochastic deviaton from zero in the CymR-Halox2/TetR-GFPx2 dataset, due to the lower number of measurements included.

CymR-Halox2/TetR-GFPx2: (34170)

CymR-GFP/TetR-tdTom: (91962)

As an alternative method for chromatic aberration correction, we can apply a correction across the dataset to force the distances in X, Y, and Z to be centered at zero. Applying this correction to our data does not change any of the findings reported in the manuscript.

How did authors align color channels?

No additional alignment of color channels was performed except the correction applied for chromatic aberration. The channels used to calculate cuO and tetO positions (green and red/far-red) use the same filter and dichroic mirror and so have identical optical paths. Distinct fluorescence images are captured by toggling the laser used as input.

How did authors determine 3D positions? Was it PSF-fitting? If so, what was the PSF-model? Did they enforce symmetric XY PSF or allow asymmetric XY-PSF? Did they do MLE or LS fitting?

We used the Laplacian of Gaussian detector available from TrackMate with sub-pixel localization enabled. From TrackMate v3.7.0: This detector applies a LoG (Laplacian of Gaussian) filter to the image, with a σ suited to the blob estimated size. Calculations are made in Fourier space. The maxima in the filtered image are searched for, and maxima too close from each other are suppressed. A quadratic fitting scheme allows to do sub-pixel localization.

What were the settings used in TrackMate? Were gaps allowed?

We did allow gap closing of no more than 3 frames in the assembly of tracks. We have added this and other details regarding the TrackMate settings used in the Materials and methods section Image Analysis.

[Editors' note: further revisions were requested prior to acceptance, as described below.]

The manuscript is acceptable but there are some remaining issues that need to be addressed. In short, the reviewers would like to see some additional comments regarding the lateral resolution and the inclusion of some of the raw data.

We are pleased that the reviewer was impressed with the precision of our measurements. We took a lot of effort in both the cell line generation (where the appropriate level of repressor expression was important in getting the best signal-to-noise), during the imaging (where the rapid piezo-controlled z drive and triggered acquisition gave us the needed speed for a rapid 3D acquisition for accurate measurement of loci position), and in our data analysis platform—where the same Trackmate algorithm that is used for single-particle tracking (Tinevez et al., 2017) performed well for assaying loci position. We have explained each of these points in our Materials and methods and also include several raw and denoised data stacks (deposited in the Zenodo data repository doi: 10.5281/zenodo.2658814 https://zenodo.org/record/2658814#.XNDLAhNKjyw) so the readers can get a sense for the quality of our raw data and so can use this to replicate our full analysis pipeline.

We chose not to specifically discuss why our data is better than what is typically achieved by others in the main text, because although the quality of our analysis may be better than some other papers in the field, it is certainly not unprecedented for tracking chromosomal loci, where several previous papers have achieved as good or better precision as our work:

40 nm precision in 3-dimensions for lacO:lacI-GFP, Marshall et al., 1997

15-30 nm precision in x, y, and z for lacO:lacI-GFP and tetO:tetR-mCheery, PMID: 29296501

25 nm precision in 3-dimensions for lacO:lacI-GFP, PMID: 27410730

As with these other reports, we were able to achieve this precision despite using standard fluorescence acquisition by using the Gaussian distribution of photons from each label collected by the camera to determine the center of fluorescence with subpixel precision. Thus, we are doing super-resolution localization in a manner similar to that performed for STORM/PALM, where precisions as high as 5 nm have been reported (PMID: 16028892), and 10-75 nm resolution in XY and Z is common for single-molecules (PMID: 26546293).